# MeCP2 in cholinergic interneurons of nucleus accumbens regulates fear learning

**Ying Zhang[1†‡], Yi Zhu[1†], Shu-Xia Cao[2†], Peng Sun[1], Jian-Ming Yang[1], Yan-Fang Xia[1], Shi-Ze Xie[1], Xiao-Dan Yu[1], Jia-Yu Fu[1], Chen-Jie Shen[1], Hai-Yang He[1], Hao-Qi Pan[1], Xiao-Juan Chen[1], Hao Wang[1], Xiao-Ming Li[1,3]\***

[1]Center for Neuroscience and Department of Neurology of Second Affiliated Hospital, Zhejiang University School of Medicine, Hangzhou, China; [2]Sir Run Run Shaw Hospital, Zhejiang University School of Medicine, Hangzhou, China; [3]NHC and CAMS Key Laboratory of Medical Neurobiology, Center for Brain Science and Brain-Inspired Intelligence, Guangdong-Hong Kong-Macao Greater Bay Area, Joint Institute for Genetics and Genome Medicine between Zhejiang University and University of Toronto, Toronto, Canada

**\*For correspondence:**
lixm@zju.edu.cn

[†]These authors contributed equally to this work

**Present address:** [‡]McGovern Institute for Brain Research, Massachusetts Institute of Technology, Cambridge, United States

**Competing interests:** The authors declare that no competing interests exist.

**Abstract** Methyl-CpG-binding protein 2 (MeCP2) encoded by the *MECP2* gene is a transcriptional regulator whose mutations cause Rett syndrome (RTT). *Mecp2*-deficient mice show fear regulation impairment; however, the cellular and molecular mechanisms underlying this abnormal behavior are largely uncharacterized. Here, we showed that *Mecp2* gene deficiency in cholinergic interneurons of the nucleus accumbens (NAc) dramatically impaired fear learning. We further found that spontaneous activity of cholinergic interneurons in *Mecp2*-deficient mice decreased, mediated by enhanced inhibitory transmission via α2-containing GABA$_A$ receptors. With MeCP2 restoration, opto- and chemo-genetic activation, and RNA interference in ChAT-expressing interneurons of the NAc, impaired fear retrieval was rescued. Taken together, these results reveal a previously unknown role of MeCP2 in NAc cholinergic interneurons in fear regulation, suggesting that modulation of neurons in the NAc may ameliorate fear-related disorders.

## Introduction

MeCP2 is a nuclear protein involved in the transcriptional repression of target genes (*Chahrour and Zoghbi, 2007*; *Lyst and Bird, 2015*; *Qiu, 2018*). Mice lacking *Mecp2* can reproduce various Rett syndrome (RTT) phenotypes, including motor dysfunction, anxiety behavior, abnormal social behavior as well as impaired learning and memory (*Shahbazian et al., 2002*; *Gemelli et al., 2006*; *Adachi et al., 2009*; *Ito-Ishida et al., 2015*; *Chao et al., 2010*; *Fyffe et al., 2008*; *Chen et al., 2001*; *Guy et al., 2001*; *Moretti et al., 2006*). However, the mechanisms underlying learning and memory deficits remain poorly characterized. Given the important role of cholinergic neurons in learning and memory (*Jiang et al., 2016*; *Brown et al., 2012*; *Picciotto et al., 2012*; *Letzkus et al., 2011*; *Likhtik and Johansen, 2019*; *Hasselmo, 2006*; *Yu and Dayan, 2005*; *Aitta-Aho et al., 2018*), we proposed that conditional deletion of *Mecp2* from cholinergic neurons may result in fear encoding dysfunction.

There are two primary sources of acetylcholine in the brain, that is, projection neurons located in the pedunculopontine, medial habenula, and basal forebrain (BF), and local interneurons located in the striatum and NAc (*Aitta-Aho et al., 2018*; *Floresco, 2015*; *Kreitzer, 2009*; *Ren et al., 2011*). Our previous study suggested that cholinergic projection neurons of the BF may contribute to impaired social interaction, decreased anxiety- and depression-like behaviors as well as elevated epilepsy susceptibility in *Mecp2*-deficient mice, with no effect on hippocampus-dependent learning and memory (*Zhang et al., 2016b*). Although MeCP2 is highly expressed in NAc neurons (*Deng et al.,*

*2010*), relatively little is known about how MeCP2 regulates NAc cholinergic interneuron function. Despite the paucity of cholinergic interneurons in the NAc, acetylcholine signaling is directly required for the modulation of local circuit activity and associative learning (*Cardinal et al., 2002*; *Luchicchi et al., 2014*; *Mamaligas and Ford, 2016*; *Picciotto et al., 2012*; *Shen et al., 2005*; *Witten et al., 2010*; *Brown et al., 2012*; *Hikida et al., 2016*; *Lee et al., 2016*). Additionally, clinical research has implicated the NAc as one of the affected brain regions in RTT patients (*Reiss et al., 1993*; *Subramaniam et al., 1997*). Considering these findings, investigating the potential role of MeCP2 in regulating the function of NAc cholinergic interneurons in associative learning and memory would be of interest.

In this study, we achieved MeCP2 deletion primarily in cholinergic neurons by crossing *Chat^Cre* mice with mice carrying the loxp-flanked allele (*Mecp2^{flox/-}*). We found that MeCP2 in NAc cholinergic interneurons participate in fear regulation. We also revealed that NAc cholinergic interneurons bi-directionally regulate fear learning. We further observed a robust reduction in NAc cholinergic interneuron activity in *Mecp2*-deficient mice, which could be attributed to elevated α2-containing GABA$_A$ receptors. With MeCP2 restoration, opto- and chemo-genetic activation, and RNA interference in NAc cholinergic interneurons, the impaired fear retrieval could be rescued. Taken together, we explored the cellular and molecular mechanisms underlying learning and memory deficits in a mouse model of RTT and discovered a previously unknown role of the NAc in regulating fear learning.

## Results

### Chat-Mecp2$^{-/y}$ mice exhibited deficits in fear learning

We first deleted *Mecp2* specifically from cholinergic neurons by crossing *Chat^Cre* mice with mice carrying the loxp-flanked allele (*Mecp2^{-/-}*), hereafter referred to as Chat-Mecp2$^{-/y}$ mice. Fluorescent immunohistochemistry results demonstrated a remarkable reduction in the expression of MeCP2 in the cholinergic neurons of Chat-Mecp2$^{-/y}$ mice (*Figure 1A–E*), thus confirming the efficiency of *Mecp2* deletion. We then examined Chat-Mecp2$^{-/y}$ mice under fear conditioning. A train of tone (30 s, 87 db) was used as conditioned stimulus (CS) and the electric footshock (0.5 mA, 1.5 ms) was used as unconditioned stimulus (US). CS was paired with US in a neural context for four trials in the training procedure. After pairing, mice were exposed to CS presented alone (context or tone) on the second day. The Chat-Mecp2$^{-/y}$ mice displayed increased freezing across trials in the training day (*Figure 1F*). However, compared to the other three control groups, freezing time of Chat-Mecp2$^{-/y}$ mice was significantly decreased on the last trial, suggesting an impairment in fear learning (*Figure 1—figure supplement 1A*). In the context and tone retrieval test, Chat-Mecp2$^{-/y}$ mice also showed decreased freezing time (*Figure 1G,H*). The difference in retrieval test with Chat-Mecp2$^{-/y}$ mice disappeared after normalizing to the last trial in the training day, further supporting the fear learning deficits (*Figure 1—figure supplement 1B,C*). The Chat-Mecp2$^{-/y}$ mice exhibited normal ability in pain detection (*Figure 1—figure supplement 1D*). We also explored whether Chat-Mecp2$^{-/y}$ mice showed spontaneous immobility behavior. The Chat-Mecp2$^{-/y}$ mice demonstrated normal locomotor activity without significant differences in immobility time in the open field tests (*Figure 1—figure supplement 1E,F*). Furthermore, these mice showed no deficit in the Y maze (*Figure 1—figure supplement 1G*) or Morris water maze tests (*Zhang et al., 2016b*), suggesting normal working and spatial memory ability.

These results indicate that conditional deletion of *Mecp2* from cholinergic neurons results in impaired fear learning.

### Restoration of *Mecp2* in NAc cholinergic interneurons rescued fear learning in Chat-Mecp2$^{-/y}$ mice

To unravel which population of cholinergic neurons contributes to the behavioral features observed in Chat-Mecp2$^{-/y}$ mice, we injected the AAV-FLEX-Mecp2-GFP (AAV/Mecp2) vector into different brain areas to specifically restore *Mecp2* in cholinergic neurons, as described previously (*Zhang et al., 2016b*). We employed immunohistochemistry to confirm the recombination efficiency of the AAV/Mecp2 virus (*Figure 2A*) and revealed that ~73% of ChAT positive neurons were infected by the AAV virus and over 85% of GFP-labeled neurons were stained positively for MeCP2 in the

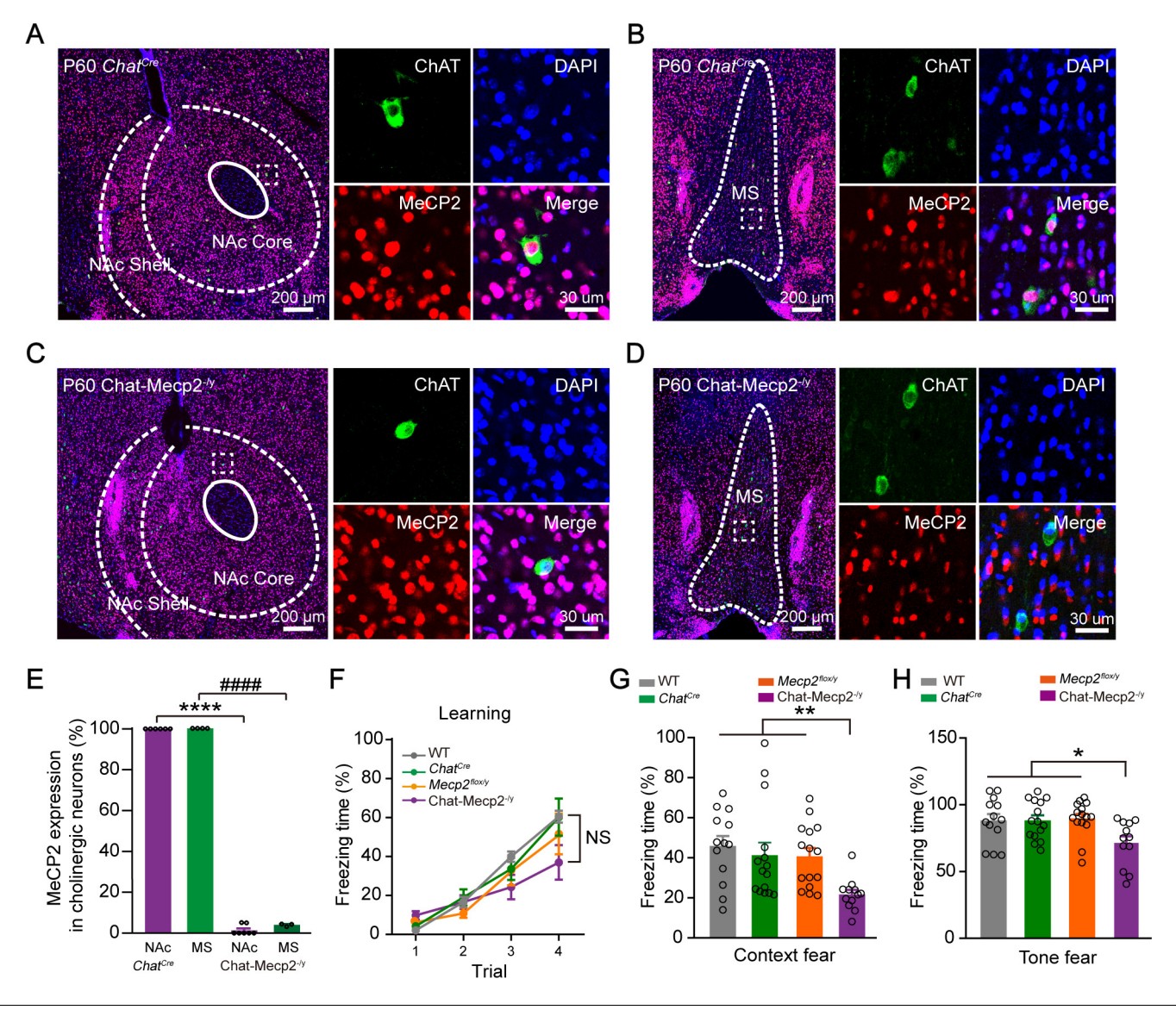

**Figure 1.** Depletion of MeCP2 in Cholinergic Neurons Resulted in Impaired Fear Regulation. (**A, B**) Fluorescence images showing NAc (**A**) and MS sections (**B**) in *Chat*^Cre^ mice stained for ChAT, MeCP2, and DAPI. (**C, D**) Fluorescence images showing NAc (**C**) and MS sections (**D**) in Chat-Mecp2$^{-/y}$ mice stained for ChAT, MeCP2, and DAPI. (**E**) Percentage of MeCP2 expression in NAc or MS of *Chat*^Cre^ or Chat-Mecp2$^{-/y}$ mice. *P*-values were calculated by two-tailed unpaired *t*-test. $t$ = 95.73, df = 11, p<0.0001 for NAc. $t$ = 281.8, df = 5, p<0.0001 for MS. n = 6 sections from four mice (NAc in *Chat*^Cre^ mice), 4 sections from four mice (MS in *Chat*^Cre^ mice), 7 sections from three mice (NAc in Chat-Mecp2$^{-/y}$ mice), 3 sections from three mice (MS in Chat-Mecp2$^{-/y}$ mice). (**F**) No significant change was detected in fear learning. *P*-values were calculated by two-way analysis of variance (ANOVA) with Tukey's multiple comparisons test. $F_{(9, 153)}$=1.895, p=0.0565. (**G, H**) Chat-Mecp2$^{-/y}$ mice showed decreased percentage of time freezing in context retrieval (**G**) and tone retrieval (**H**). *P*-values were calculated by one-way ANOVA with Bonferroni's multiple comparisons test. $F_{(3, 51)}$=4.445, p=0.0075 for (**G**). $F_{(3, 51)}$=3.941, p=0.0132 for (**H**). n = 13 (WT), 15 (*Chat*^Cre^), 15 (*Mecp2*^flox/y^), 12 (Chat-Mecp2$^{-/y}$) mice. NAc: nucleus accumbens. MS: medial septum. WT: wild type mice. ChAT: Choline acetyltransferase, as a marker for cholinergic neurons. Data are means ± SEM. *p<0.05, **p<0.01, ***p<0.001, ****p<0.0001, ####p<0.0001. NS means no significance.

The online version of this article includes the following source data and figure supplement(s) for figure 1:

**Source data 1.** Statistical reporting of *Figure 1*.

**Figure supplement 1.** Chat-Mecp2$^{-/y}$ Mice Showed Normal Ability in Pain Perception, Movement Ability, and Working Memory, Related to *Figure 1*.

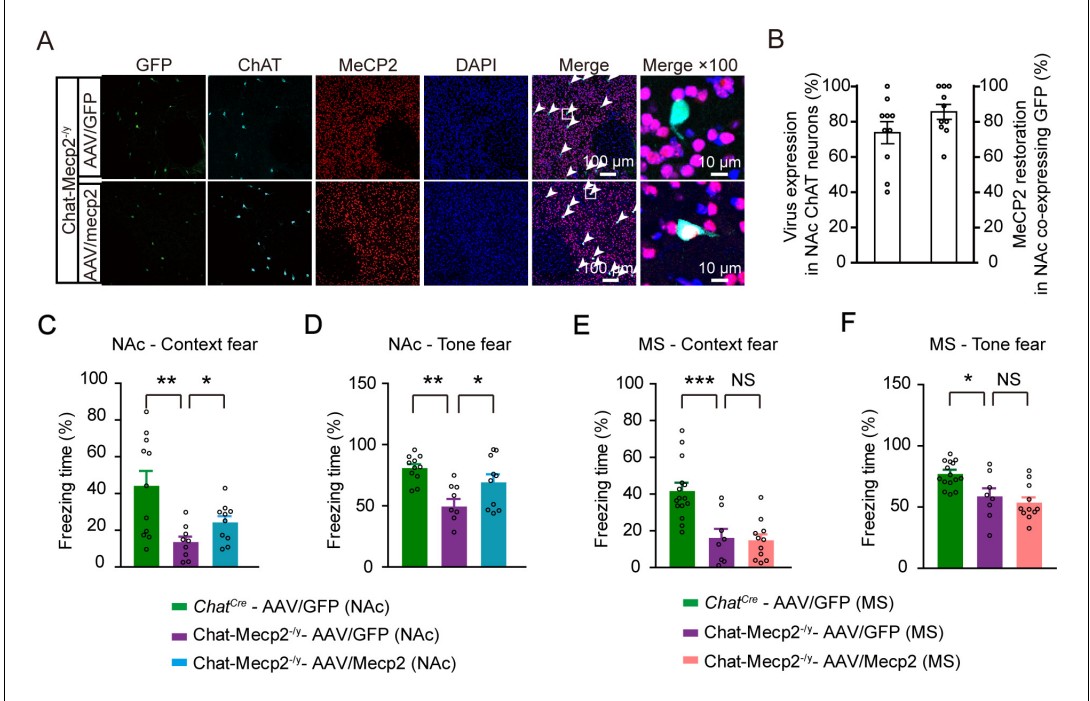

**Figure 2.** Restoration of MeCP2 in NAc rather than Cholinergic Neurons of Medial Septum Rescued Fear Deficits in Chat-Mecp2$^{-/y}$ Mice. (**A**) Representative fluorescence images showing NAc sections of Chat-Mecp2$^{-/y}$-AAV/GFP and Chat-Mecp2$^{-/y}$-AAV/Mecp2 stained for DAPI, MeCP2, and ChAT. (**B**) Percentage of ChAT-positive cells in NAc co-expressing overexpression virus (left) and efficiency of MeCP2 restoration in GFP-positive neurons (right) in Chat-Mecp2$^{-/y}$-AAV/Mecp2 mice. n = 10 mice. (**C–F**) Fear conditioning test was conducted three weeks after microinjection in NAc (**C**, **D**) or MS (**E**, **F**). *P*-values were calculated by one-way analysis of variance (ANOVA) with Bonferroni's multiple comparisons test. $F_{(2, 27)}=7.304$, p=0.0029 for (**C**). $F_{(2, 27)}=8.469$, p=0.0014 for (**D**). $F_{(2, 31)}=14.52$, p<0.0001 for (**E**). $F_{(2, 31)}=8.261$, p=0.0013 for (**F**). n = 11 (*Chat$^{Cre}$*-AAV/GFP in NAc), 9 (Chat-Mecp2$^{-/y}$-AAV/GFP in NAc), 10 (Chat-Mecp2$^{-/y}$-AAV/Mecp2 in NAc); 15 (*Chat$^{Cre}$*-AAV/GFP in MS), 8 (Chat-Mecp2$^{-/y}$-AAV/GFP in MS), 11 (Chat-Mecp2$^{-/y}$-AAV/Mecp2 in MS). Data are means ± SEM. *p<0.05, **p<0.01, ***p<0.001, NS means no significance.

The online version of this article includes the following source data and figure supplement(s) for figure 2:

**Source data 1.** Statistical reporting of *Figure 2*.
**Figure supplement 1.** Systemic Delivery of Various AAV Vectors into NAc, Related to *Figures 2*, *4*, *5*.
**Figure supplement 2.** Restoration of MeCP2 in NAc rather than Cholinergic neurons of MS Rescued Fear deficit at the last trial of learning, Related to *Figure 2*.

NAc (*Figure 2B*; *Figure 2—figure supplement 1A,B*). To test the behavioral effects of MeCP2 restoration in cholinergic neurons, we performed fear conditioning tests. It appeared that fear learning deficit of Chat-Mecp2$^{-/y}$ mice was rescued towards the end of training with MeCP2 restoration in the cholinergic neurons of NAc, but not medial septum (MS) (*Figure 2—figure supplement 2*). Consistently, in the context and tone fear retrieval tests, although the freezing level was not comparable to that of the AAV/GFP-injected *Chat$^{Cre}$* mice, Chat-Mecp2$^{-/y}$ mice injected with the AAV/Mecp2 virus showed a significant enhancement (*Figure 2C,D*). In contrast, MeCP2 restoration in the cholinergic neurons of MS, which has been reported to mediate sensory induced aversion (*Zhang et al., 2018*), had no effect on fear regulation in Chat-Mecp2$^{-/y}$ mice (*Figure 2E,F*).

Taken together, these regional restoration results indicate that impaired fear learning in Chat-Mecp2$^{-/y}$ mice arises, at least in part, from the lack of MeCP2 expression in NAc cholinergic interneurons.

## Spontaneous firing of cholinergic interneurons decreased via elevated expression of GABA$_A$ α2 receptors in Chat-Mecp2$^{-/y}$ mice

We next sought to uncover the cellular mechanisms underlying impaired fear learning in Chat-Mecp2$^{-/y}$ mice. First, we measured the spontaneous firing frequency of cholinergic interneurons in brain slices, as NAc cholinergic interneurons are considered tonic firing neurons (*Zahar and Morris,*

*2014*). The spontaneous spiking rate of NAc cholinergic interneurons was significantly decreased in Chat-Mecp2$^{-/y}$ mice in both cell-attached (*Figure 3A,B*) and whole-cell configurations (*Figure 3C,D*). We also evaluated the spontaneous firing rate and found a significant increase in Chat-Mecp2$^{-/y}$ mice with MeCP2/AAV expression compared with GFP/AAV expression (*Figure 3E,F*). According to these results, we found that the fear learning deficits in Chat-Mecp2$^{-/y}$ mice could be attributed to decreased tonic firing in NAc cholinergic interneurons.

Changes in spontaneous activity can be attributed to changes in either intrinsic properties or synaptic inputs (*Cheng et al., 2019*; *Brown et al., 2012*). Thus, we first measured the intrinsic properties of the cholinergic interneurons in the NAc and found no significant differences between Chat-Mecp2$^{-/y}$ and control mice (*Figure 3—figure supplement 1*). To examine the synaptic inputs of the cholinergic interneurons, we measured miniature inhibitory postsynaptic currents (mIPSCs). Although the frequency of mIPSCs was not altered in the NAc cholinergic interneurons of Chat-Mecp2$^{-/y}$ mice, their amplitude was significantly increased compared to that in control mice, indicating abnormal changes in postsynaptic connections in the cholinergic interneurons (*Figure 3G,H*; *Figure 3—figure supplement 2A*). Consistent with previous studies (*Bennett and Wilson, 1999*), our whole-cell voltage-clamp recordings showed that miniature excitatory postsynaptic currents (mEPSCs) were rarely recorded in the cholinergic interneurons (mEPSC frequency <0.1 Hz, data not shown).

The GABA$_A$ receptor is a heteropentameric ligand-gated chloride channel that mediates major inhibitory transmission in the brain (*Farrant and Nusser, 2005*). Receptors with different subunits are distributed in different neurons and play distinct roles in current responses (*Lavoie et al., 1997*). Surprisingly, we found that approximately 70% of ChAT-positive neurons expressed detectable levels of α2 subunits in the NAc (*Figure 3I*). In contrast, there was little expression of α1 subunits in the NAc cholinergic interneurons (*Figure 3—figure supplement 2B*). To determine whether the synaptic phenotypes in Chat-Mecp2$^{-/y}$ mice could be ascribed to aberrant GABA$_A$ receptor expression, we assessed the expression of α subunits in the NAc by western blot analysis. We found that α2 was increased in Chat-Mecp2$^{-/y}$ mice but α1 was unchanged (*Figure 3J*; *Figure 3—figure supplement 2C*). Furthermore, immunohistochemistry results showed a significant increase in α2 expression specifically in the cholinergic interneurons of Chat-Mecp2$^{-/y}$ mice compared to those of *Chat*$^{Cre}$ mice (*Figure 3K*).

We next investigated the role of α2-containing GABA$_A$ receptors in regulating the spiking of cholinergic interneurons in NAc. L-838,417, an α2-subunit-seleive positive allosteric modulator of GABA$_A$ receptors, has no efficacy in α1 subtypes. Potentiating the α2-GABA$_A$ receptors with L-838,417 (2 μM) in bath solution significantly decreased the firing frequency of cholinergic interneurons in NAc slices from both control (*Figure 3L,M*) and Chat-Mecp2$^{-/y}$ mice (*Figure 3—figure supplement 2D*). Further, we also compared the effect of L-838,417 on GABAergic current and found that both the frequency and amplitude of sIPSC have been increased after treatment (*Figure 3—figure supplement 2E,F*).

Together, these results indicate that MeCP2 deficiency in cholinergic interneurons enhances inhibitory synaptic transmission, which could be ascribed to increased α2-GABA$_A$ expression.

## Activation of NAc cholinergic interneurons in Chat-Mecp2$^{-/y}$ mice rescued fear deficits

We further examined whether enhancing the spontaneous firing of NAc cholinergic interneurons is sufficient to improve freezing response in Chat-Mecp2$^{-/y}$ mice. We used AAV-DIO-hM3Dq-mCherry (AAV/hM3Dq), an excitatory designer receptor exclusively activated by designer drugs (*Figure 4A*). Electrophysiological analysis of NAc slices confirmed that 1 μM clozapine-N-oxide (CNO) activated spiking in hM3Dq-expressed NAc cholinergic interneurons (*Figure 4—figure supplement 1A,B*). We then examined the effect of CNO (1 mg/kg) administration in Chat-Mecp2$^{-/y}$ mice with AAV/hM3Dq delivered to NAc cholinergic interneurons. Activation of NAc cholinergic interneurons significantly increased the freezing level of Chat-Mecp2$^{-/y}$ mice in the training process (*Figure 4—figure supplement 1C*), as well as the context and tone fear memory retrieval process (*Figure 4B,C*). We manipulated the NAc cholinergic interneurons only in fear learning, with the fear memory test conducted without additional manipulations. To confirm these behavioral results, we selectively activated cholinergic interneurons by infusing AAV-DIO-ChR2-EYFP (AAV/ChR2) into the NAc of Chat-Mecp2$^{-/y}$ mice and inserted an optical fiber above it bilaterally (*Figure 4D*). Electrophysiological recordings of NAc cholinergic interneurons reliably revealed one-to-one, photo-stimulation locked action potential

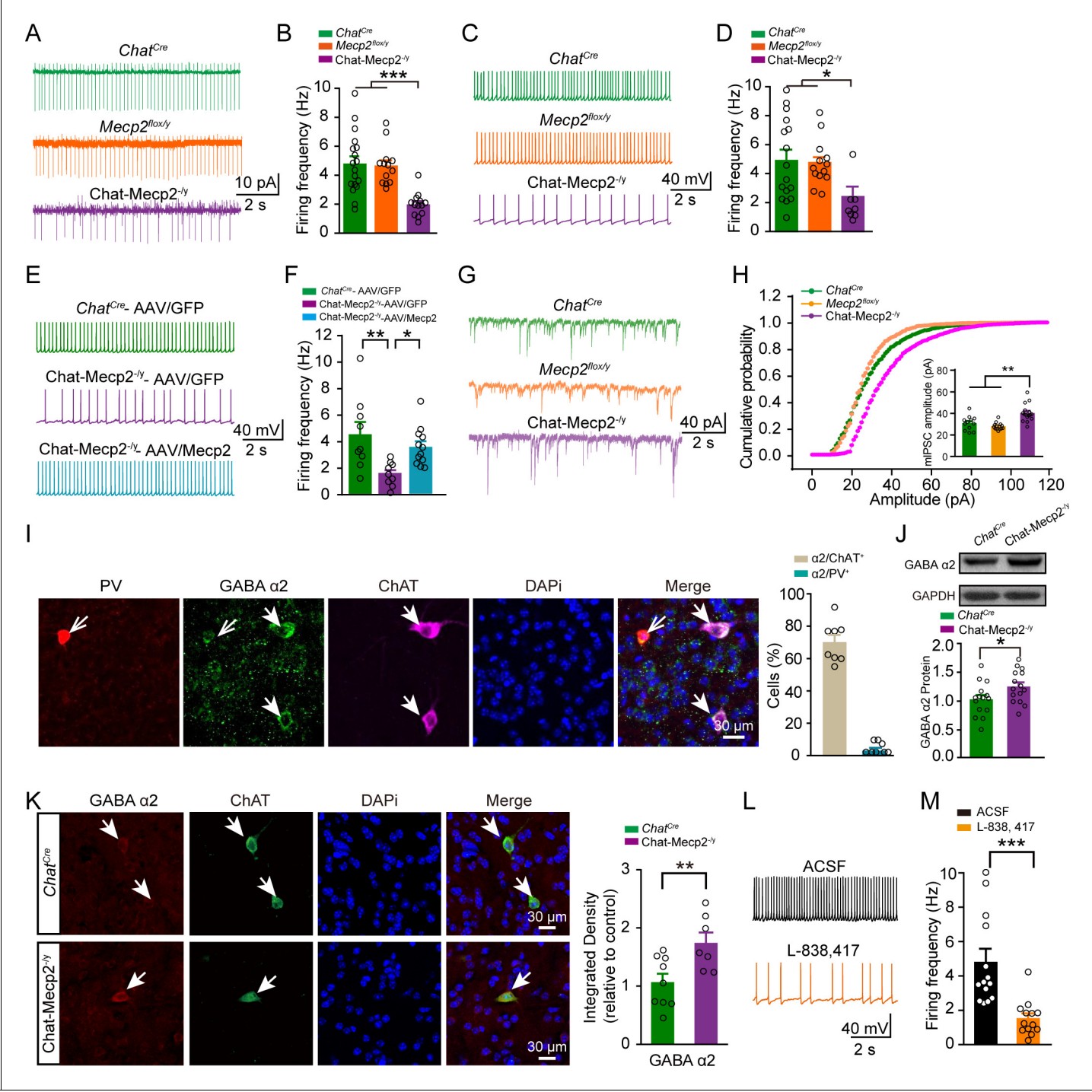

**Figure 3.** Spontaneous Firing of Cholinergic Interneurons Decreased via Elevated Expression of GABA$_A$ α2 Receptors in Chat-Mecp2$^{-/y}$ Mice. (**A, B**) Example traces of spontaneous spiking (**A**) or averaged frequency (**B**) measured in a cell-attached configuration from NAc cholinergic interneurons. *P*-values were calculated by one-way analysis of variance (ANOVA) with Bonferroni's multiple comparisons test. $F_{(2, 41)}=13.93$, $p<0.0001$. n = 18 neurons from three mice (*Chat$^{Cre}$*), 13 neurons from three mice (*Mecp2$^{flox/y}$*), 13 neurons from five mice (Chat-Mecp2$^{-/y}$). (**C, D**) Example traces of spontaneous spiking (**C**) or averaged frequency (**D**) measured in a whole-cell configuration from NAc cholinergic interneurons. *P*-values were calculated by one-way ANOVA with Bonferroni's multiple comparisons test. $F_{(2, 34)}=5.336$, $p=0.0096$. n = 16 neurons from three mice (*Chat$^{Cre}$*), 13 neurons from four mice (*Mecp2$^{flox/y}$*), 8 neurons from five mice (Chat-Mecp2$^{-/y}$). (**E, F**) Example traces of spontaneous spiking (**E**) or averaged frequency (**F**) measured in a whole-cell configuration from NAc neurons infected by AAV-GFP or AAV/Mecp2 for indicated genotype. Data are means ± SEM. *P*-values were calculated by one-way ANOVA with Tukey's multiple comparisons test. $F_{(2, 27)}=6.201$, $p=0.0061$. n = 9 neurons from three mice (*Chat$^{Cre}$*-AAV/GFP), 9 neurons from three mice (Chat-Mecp2$^{-/y}$-AAV/GFP), 12 neurons from three mice (Chat-Mecp2$^{-/y}$-AAV/Mecp2). (**G, H**) Example mIPSC traces (**G**)

*Figure 3 continued on next page*

*Figure 3 continued*

measured in whole-cell configuration from NAc cholinergic interneurons. (H) Cumulative distribution of mIPSC amplitude of NAc cholinergic interneurons. *P*-values were calculated by one-way ANOVA with Bonferroni's multiple comparisons test. F (2, 39)=15.46, p<0.0001. n = 12 neurons from four mice (*Chat$^{Cre}$*), 14 neurons from four mice (*Mecp2$^{flox/y}$*), 16 neurons from four mice (Chat-Mecp2$^{-/y}$). (I) Confocal images showing NAc sections stained for PV, ChAT, GABA$_A$ α2 receptor, and DAPI. Bar graph showing percentage of ChAT or PV-positive cells co-expressing GABA$_A$ α2 receptor in wild-type (WT) mice. n = 8 sections from two mice. (J) Immunoblotting of GABA$_A$ α2 receptor in NAc extracts prepared from *Chat$^{Cre}$* and Chat-Mecp2$^{-/y}$ mice. Each lane was loaded with 40 μg of protein, with GAPDH as loading control, and normalized to *Chat$^{Cre}$* levels. *P*-values were calculated by two-tailed unpaired *t*-test. t = 2.153, df = 28, p=0.0401. n = 15 mice per group. (K) Representative images of NAc slices from *Chat$^{Cre}$* and ChAT-Mecp2$^{-/y}$ mice. Statistically integrated immunofluorescence data normalized to *Chat$^{Cre}$* levels. *P*-values were calculated by two-tailed unpaired *t*-test. t = 2.986, df = 14, p=0.0098. n = 7–9 sections from two mice per group. (L, M) Example traces (L) and statistical results (M) of spontaneous spiking recorded from cholinergic interneurons in NAc of different groups. *P*-values were calculated by two-tailed unpaired *t*-test. t = 3.977, df = 25, p=0.0005. n = 14 neurons from four mice for ACSF, n = 13 neurons from four mice for L-838,417. L-838,417: an α2-subunit-seleceive selective positive allosteric modulator of GABA$_A$ receptors. Data are means ± SEM. *p<0.05, **p<0.01, ***p<0.001, NS means no significance.

The online version of this article includes the following source data and figure supplement(s) for figure 3:

**Source data 1.** Statistical reporting of *Figure 3*.

**Figure supplement 1.** Chat-Mecp2$^{-/y}$ Mice Exhibited No Statistical Difference in Intrinsic Properties in NAc Cholinergic Interneurons, Related to *Figure 3*.

**Figure supplement 2.** Depletion of MeCP2 Did Not Affect mIPSC Frequency of Cholinergic Interneurons, with Limited Expression of GABA$_A$ Receptor α1 Subunits in Cholinergic Interneurons within NAc, Related to *Figure 3*.

firing at frequencies up to 10 Hz (*Figure 4—figure supplement 1D*). We found that optogenetic activation of cholinergic interneurons (8 Hz) only during fear learning also rescued fear deficits in Chat-Mecp2$^{-/y}$ mice (*Figure 4E,F* and *Figure 4—figure supplement 1E*). We also confirmed that activation of NAc cholinergic neurons had no effect on freezing in the absence of shock delivery (*Figure 4—figure supplement 1F–H*).

These results indicate that activation of the decreased activity in NAc cholinergic interneurons could rescue the fear learning deficits in Chat-Mecp2$^{-/y}$ mice.

## Inhibition of NAc cholinergic interneurons resulted in fear deficits

To investigate if NAc cholinergic neurons play a causal role in fear encoding, we selectively inhibited NAc cholinergic interneurons in vivo by delivering the AAV-DIO-hM4Di-mCherry vector into the NAc of *Chat$^{Cre}$* mice (*Figure 5A*). Application of CNO before fear conditioning significantly decreased context and tone fear responses during the retrieval process (*Figure 5B,C*). As a key region of emotional valence encoding, the NAc is composed of two subregions, that is the shell and core, which have different anatomical structure and function (*Castro and Bruchas, 2019*; *de Jong et al., 2019*). We wondered if these two subregions play different roles in fear regulation. As such, we used optogenetic manipulation to inhibit the activity of cholinergic interneurons in the NAc precisely. We injected AAV-FLEX-NpHR-EYFP (AAV/NpHR), an AAV-DIO vector expressing halorhodopsin fused with EYFP, and inserted optical fibers above the NAc core, the medial shell, and ventral medial shell bilaterally into *Chat$^{Cre}$* mice separately (*Figure 5—figure supplement 1*). Inhibition of cholinergic neurons in core or shell has similar effect on context/tone fear. However, if we only inhibit cholinergic neurons in NAc medial shell, only tone fear was impaired, with context fear intact, suggesting the heterogeneity within NAc shell (*Figure 5D,E*).

Thus, the above findings imply that inhibition of NAc cholinergic interneurons results in fear deficits.

## Down-regulation of GABA$_A$ α2 receptors in NAc cholinergic interneurons in Chat-Mecp2$^{-/y}$ mice rescued fear deficits

To determine whether down-regulation of α2-GABA$_A$ receptors in the NAc can reverse impaired fear encoding in Chat-Mecp2$^{-/y}$ mice, we knocked down α2-GABA$_A$ receptors via RNA interference (RNAi). The AAV-FLEX-siRNA-GFP (AAV/α2RNAi) virus was delivered bilaterally into the NAc of Chat-Mecp2$^{-/y}$ mice (*Figure 6A*). We first verified virus efficiency by slice recordings. In the NAc of Chat-Mecp2$^{-/y}$ mice, neurons infected by AAV/α2RNAi showed significantly decreased mIPSC amplitude compared to the scrambled controls (AAV/sham) (*Figure 6B,C*), but no changes in mIPSC frequency (*Figure 6—figure supplement 1*). Immunohistochemical analysis also revealed AAV/α2RNAi

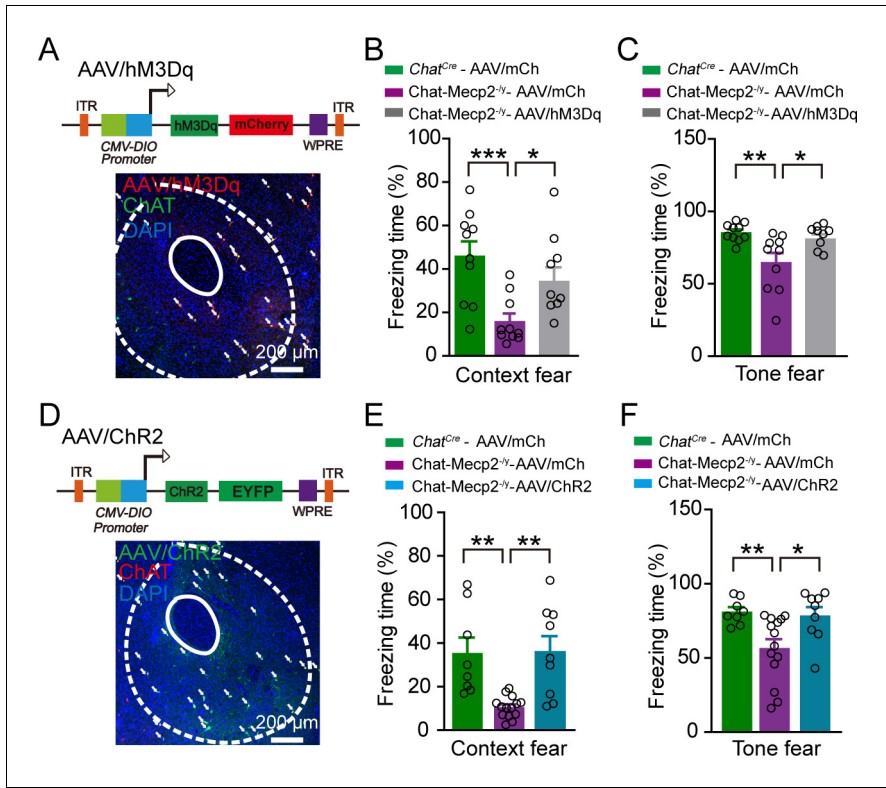

**Figure 4.** Activation of NAc Cholinergic Interneurons Rescued Fear Deficits in Chat-Mecp2$^{-/y}$ Mice. (**A**) Up: Schematic of AAV/hM3Dq viral construct. Down: Confocal image of injected NAc slice demonstrating colocalization of mCherry expression with ChAT antibody, co-stained with DAPI. (**B, C**) Fear retrieval was measured by retrieval process in fear conditioning test. *P*-values were calculated by one-way analysis of variance (ANOVA) with Bonferroni's multiple comparisons test. $F_{(2, 26)}=7.801$, p=0.0022 for (**B**). $F_{(2, 26)}=7.583$, p=0.0025 for (**C**). n = 10 (*Chat$^{Cre}$*-AAV/mCherry), 10 (Chat-Mecp2$^{-/y}$-AAV/mCherry), 9 (Chat-Mecp2$^{-/y}$-AAV/hM3Dq). (**D**) Up: Schematic of AAV/ChR2 viral construct. Down: Confocal image of AAV/ChR2 vector-injected NAc slice demonstrating colocalization of GFP expression with ChAT antibody, co-stained with DAPI. (**E, F**) Fear was measured by retrieval process in fear conditioning test. *P*-values were calculated by one-way ANOVA with Bonferroni's multiple comparisons test. $F_{(2, 28)}=10.50$, p=0.0004 for (**E**). $F_{(2, 28)}=6.356$, p=0.0053 for (**F**). n = 8 (*Chat$^{Cre}$*-AAV/mCherry), 14 (Chat-Mecp2$^{-/y}$-AAV/mCherry), 9 (Chat-Mecp2$^{-/y}$-AAV/ChR2). Data are means ± SEM. *p<0.05, **p<0.01, ***p<0.001. (See also *Figure 2—figure supplement 1*).

The online version of this article includes the following source data and figure supplement(s) for figure 4:

**Source data 1.** Statistical reporting of *Figure 4*.

**Figure supplement 1.** Activation of NAc Cholinergic Interneurons Rescued Fear Deficits in Chat-Mecp2$^{-/y}$ Mice, Related to *Figure 4*.

---

specific infection in the cholinergic interneurons of the NAc and ~50% knockdown efficiency of α2-GABA$_A$ receptors in the cholinergic interneurons (*Figure 6D,E*). Furthermore, we found that down-regulating α2-GABA$_A$ receptors in the NAc of Chat-Mecp2$^{-/y}$ mice rescued the decreased spontaneous firing rates of NAc cholinergic interneurons (*Figure 5G,H*). Functionally, the freezing response in the context and tone fear retrieval period was markedly improved in Chat-Mecp2$^{-/y}$ mice infected with AAV/α2RNAi compared with the AAV/sham group (*Figure 5I,J*). Locomotor activity was not significantly changed, suggesting that the freezing behavior was not due to motor defects (*Figure 5K*). Thus, according to these results, down-regulation of α2-GABA$_A$ receptors in the NAc of Chat-Mecp2$^{-/y}$ mice can rescue the decreased spontaneous firing rates of NAc cholinergic interneurons as well as fear deficits.

We next used mixtures of AAV/α2RNAi and AAV/NpHR to investigate whether the recuperation effect of AAV/α2RNAi in fear encoding was mediated through the spiking of cholinergic interneurons (*Figure 5F*; *Figure 2—figure supplement 1C*). We verified functionality with slice recordings

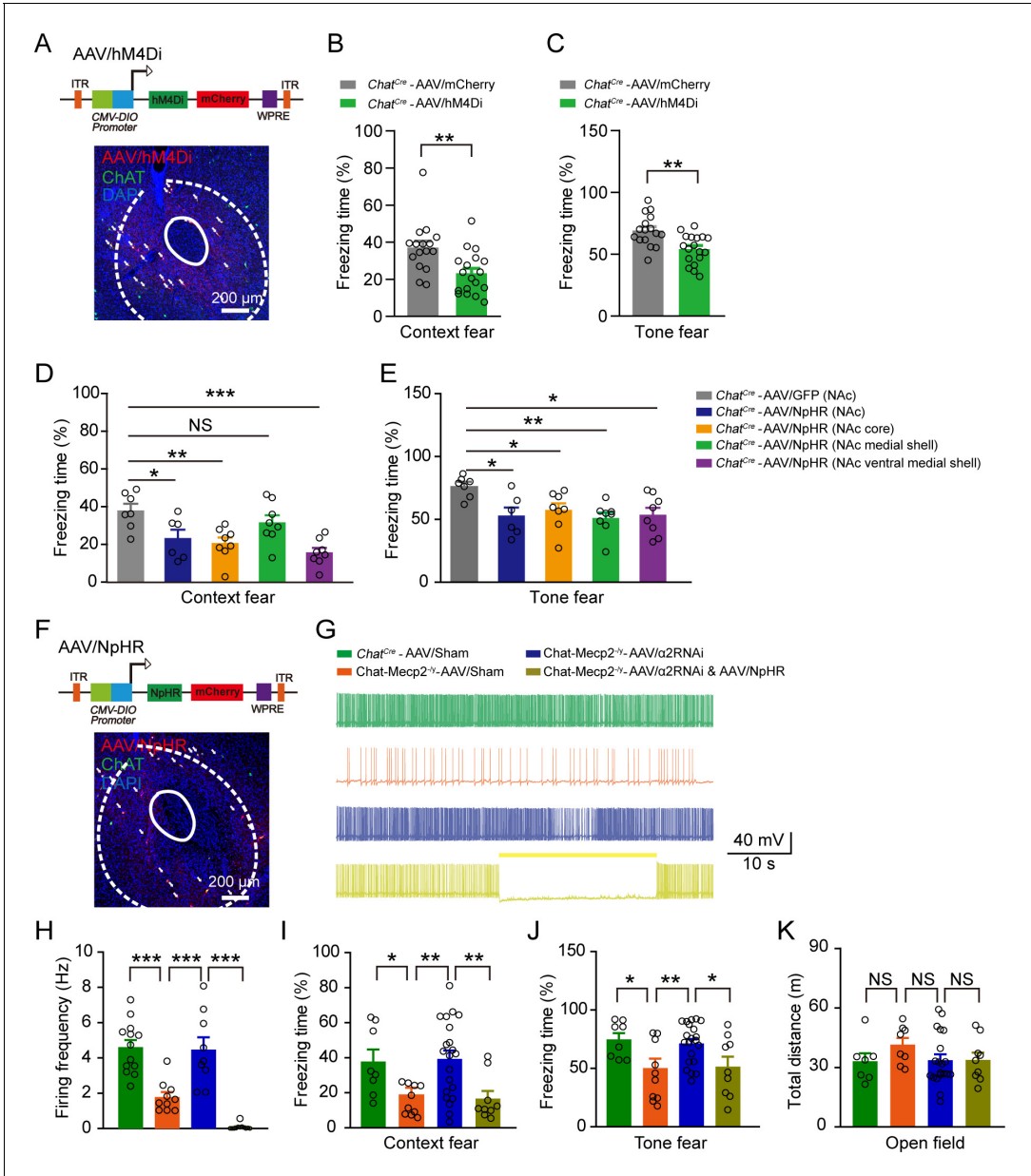

**Figure 5.** Inhibition of NAc Cholinergic Interneurons in *Chat^Cre* Mice Mimicked Fear Deficits in Chat-Mecp2^{-/y} Mice. (**A**) Schematic of AAV/hM4Di viral construct. Confocal image of injected NAc slice demonstrating colocalization of mCherry expression with ChAT antibody, co-stained with DAPI. (**B, C**) Fear retrieval was measured by retrieval process in fear conditioning test. *P*-values were calculated by two-tailed unpaired *t*-test. $t = 3.204$, df = 32, p=0.0031 for (**B**). $t = 3.572$, df = 32, p=0.0011 for (**C**). n = 16 (*Chat^Cre*-AAV/mCherry), 18 (*Chat^Cre*-AAV/hM4Di). (**D, E**) Fear retrieval was measured by retrieval process in fear conditioning test with opto-inhibition of cholinergic neurons in different NAc subregions. *P*-values were calculated by two-tailed unpaired *t*-test. $t = 2.530$, df = 11, p=0.0280 comparison of NpHR (NAc) effect in context fear. $t = 3.662$, df = 13, p=0.0029 comparison of NpHR (NAc core) effect in context fear. $t = 1.179$, df = 13, p=0.2593 comparison of NpHR (NAc medial shell) effect in context fear. $t = 5.140$, df = 13, p=0.0002 comparison of NpHR (NAc ventral medial shell) effect in context fear. $t = 3.308$, df = 11, p=0.0113 comparison of NpHR (NAc) effect in tone fear. $t = 2.600$, df = 13, p=0.0220 comparison of NpHR (NAc core) effect in tone fear. $t = 4.181$, df = 13, p=0.0011 comparison of NpHR (NAc medial shell) effect in tone fear. $t = 3.026$, df = 13, p=0.0097 comparison of NpHR (NAc ventral medial shell) effect in tone fear. n = 7 (*Chat^Cre*-AAV/GFP in NAc), 6 (*Chat^Cre*-AAV/NpHR in NAc), 8 (*Chat^Cre*-AAV/NpHR in NAc core), 8 (*Chat^Cre*-AAV/NpHR in NAc medial shell), 8 (*Chat^Cre*-AAV/NpHR in NAc ventral medial shell). (**F**) Schematic of AAV/NpHR viral construct. Confocal image of injected NAc slice demonstrating colocalization of mCherry expression with ChAT antibody, co-stained with DAPI. (**G, H**) Example traces (**G**) and statistical results (**H**) of spontaneous spiking recorded from cholinergic interneurons in NAc of different groups. *P*-values were calculated by one-way ANOVA with Newman-keuls multiple comparison test. $F_{(3, 40)}=33.06$, p<0.0001. n = 13 neurons from three mice (*Chat^Cre*-AAV/Sham), 10 neurons from three mice (Chat-Mecp2^{-/y}-AAV/Sham), 9 neurons from three mice (Chat-Mecp2^{-/y}-AAV/α2RNAi), 12 neurons from four mice (Chat-Mecp2^{-/y}-AAV/α2RNAi and AAV/NpHR). (**I, J**) Fear retrieval was measured by retrieval

*Figure 5 continued on next page*

*Figure 5 continued*

process in fear conditioning test. *P*-values were calculated by one-way ANOVA with Newman-keuls multiple comparisons test. $F_{(3, 44)}=6.066$, $p=0.0015$ in (I). $F_{(3, 44)}=5.426$, $p=0.0029$ in (J). n = 8 mice (*Chat*^Cre^-AAV/Sham), 10 mice (Chat-Mecp2^-/y^-AAV/Sham), 21 mice (Chat-Mecp2^-/y^-AAV/α2RNAi), nine mice (Chat-Mecp2^-/y^-AAV/α2RNAi and AAV/NpHR). (K) Locomotor activity (15 min) was measured in open field test. *P*-values were calculated by one-way ANOVA with Newman-keuls multiple comparisons test. $F_{(3, 40)}=0.9904$, $p=0.4071$. n = 7 (*Chat*^Cre^-AAV/Sham), 8 (Chat-Mecp2^-/y^-AAV/Sham), 20 (Chat-Mecp2^-/y^-AAV/α2RNAi), nine mice (Chat-Mecp2^-/y^-AAV/α2RNAi and AAV/NpHR). Data are means ± SEM. *p<0.05, **p<0.01, ***p<0.001, NS means no significance. (See also *Figure 2—figure supplement 1*).

The online version of this article includes the following source data and figure supplement(s) for figure 5:

**Source data 1.** Statistical reporting of *Figure 5*.

**Figure supplement 1.** Inhibition of NAc Cholinergic Interneurons in *Chat*^Cre^ Mice, Related to *Figure 5*.

and found that NpHR-mediated hyperpolarization efficiently blocked the spontaneous firing rates of cholinergic interneurons (*Figure 5G,H*). Inhibiting the spiking of cholinergic interneurons with yellow light (593 nm) only during the learning phase abolished the rescue effects of AAV/RNAi in Chat-Mecp2^-/y^ mice (*Figure 5I,J*) without affecting locomotion (*Figure 5K*). These results demonstrate that the rescue of fear learning by down-regulation of NAc α2-GABA_A receptors requires the modulation of cholinergic activity.

The above findings suggest that down-regulation of GABA_A α2 receptors in NAc cholinergic interneurons of Chat-Mecp2^-/y^ mice can rescue the decreased tonic firing rate of cholinergic interneurons and fear deficits in these mice.

Based on our evidence that α2-GABA_A receptors increase in cholinergic interneurons of Chat-Mecp2^-/y^ mice, we hypothesized that α2-GABA_A receptors are a potential target for fear learning regulation. To simulate enhanced α2-specific inhibitory transmission in Chat-Mecp2^-/y^ mice, we injected L-838,417 into the bilateral NAc of wild-type mice 30 min prior to the training period of fear conditioning only (*Figure 6F*). In the retrieval sessions, identical scenery or tone elicited significantly decreased freezing levels in the L-838,417 group (*Figure 6G,H*), suggesting that activation of α2-containing GABA_A receptors in the NAc was sufficient to dampen the spiking of cholinergic interneurons and impair fear behavior.

## Discussion

Here, we observed robust fear learning deficits and reduction in cholinergic interneuron activity in the NAc of Chat-Mecp2^-/y^ mice, which could be attributed to an elevation in α2-containing GABA_A receptors. Furthermore, with various manipulations of cholinergic interneurons in the NAc, fear learning could be regulated bi-directionally. Taken together, our study adds to growing evidence of the involvement of MeCP2 and NAc in fear regulation.

Deficiency of MeCP2 in specific neuron types or regions results in RTT-like phenotypes in mice (*Fyffe et al., 2008*; *Guy et al., 2001*; *Zhang et al., 2016b*). For example, *PV-Mecp2*^-/y^, *CAMKII-Mecp2*^-/y^, and MeCP2 region-specific deletion from the basolateral amygdala (BLA) results in anxiety-like behavior and impaired learning ability (*Adachi et al., 2009*; *Chen et al., 2001*; *Ito-Ishida et al., 2015*; *Gemelli et al., 2006*). Furthermore, *Mecp2*^308/y^ and *Viaat-Mecp2*^-/y^ mice almost recapitulate the entire spectrum of RTT features, including motor dysfunction, seizure, and stereotyped behaviors (*Chao et al., 2010*; *Shahbazian et al., 2002*; *Moretti et al., 2006*). Here, we found that dysfunction of NAc cholinergic interneurons accounted for the impaired context-dependent and cue-related fear learning in Chat-Mecp2^-/y^ mice. However, these results are inconsistent with recent findings, which suggest that Chat-Mecp2^-/y^ mice do not exhibit impaired fear conditioning (*Ballinger et al., 2019*). One possible explanation for this difference may be the use of larger footshock (2 s, 1 mA), which could render a strong fear response in over-trained mice, thereby obscuring the effect of *Mecp2* deletion in cholinergic neurons. Another possibility could be the age difference. Ballinger et al. conducted fear conditioning test when mice were 21 weeks old. However, the mice we used throughout experiments were 9 to 12 weeks. Since RTT is a neurodevelopmental disorder, there is high possibility that age difference of mice between two studies lead to distinct results. Interestingly, *Mecp2*^308/y^ mice only display context-dependent fear memory deficits (*Moretti et al., 2006*); whereas, mice with *PV-Mecp2*^-/y^, *CAMKII-Mecp2*^-/y^, and MeCP2 region-specific deletions from the BLA only

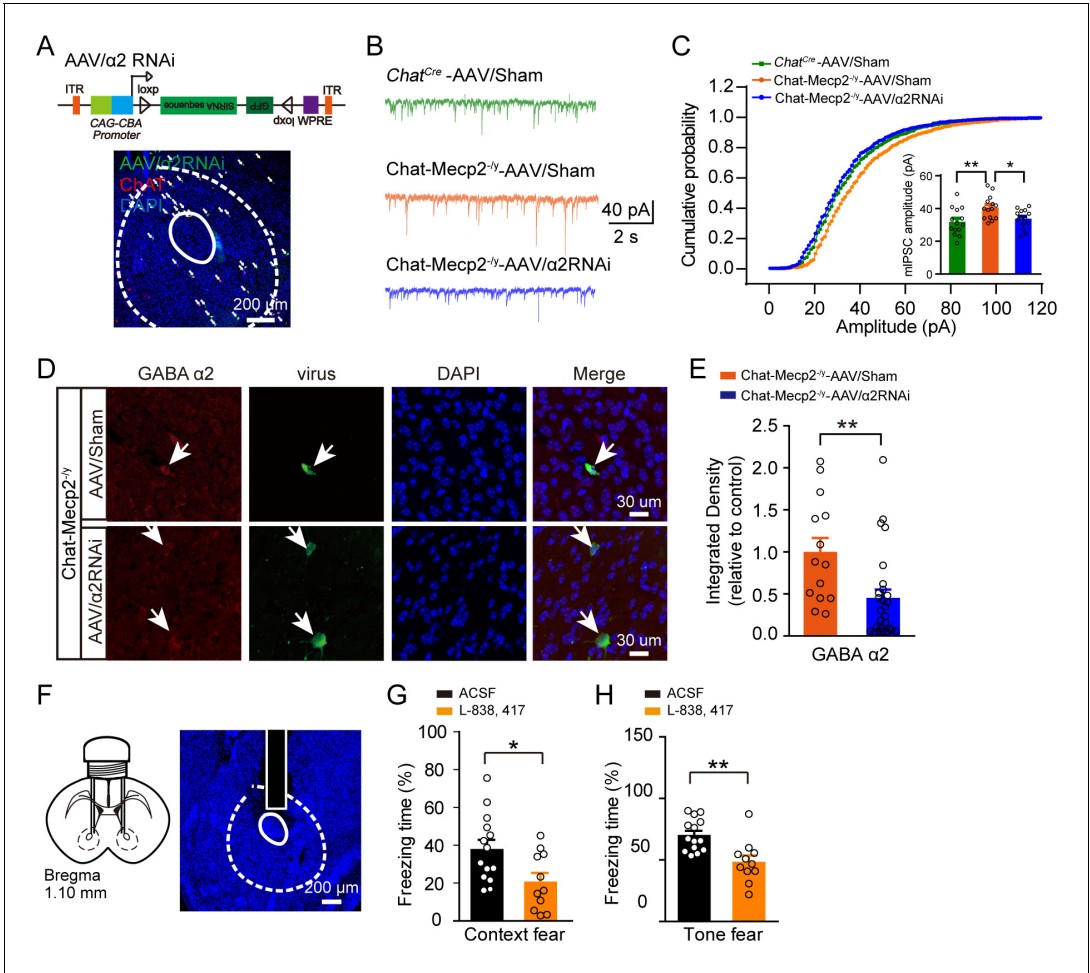

**Figure 6.** Manipulation of α2-GABA$_A$ Receptors in NAc Mimicked Fear Deficits in Chat-Mecp2$^{-/y}$ Mice. (**A**) Schematic of AAV/RNAi viral construct. Confocal image of injected NAc slice demonstrating colocalization of GFP expression with ChAT antibody, co-stained with DAPI. (**B, C**) mIPSCs of AAV/RNAi-infected NAc neurons from Chat-Mecp2$^{-/y}$ mice were rescued. (**B**) Examples of mIPSC traces. (**C**) Cumulative distribution of mIPSC amplitude of NAc neurons infected by AAV-Sham or AAV/RNAi. *P*-values were calculated by one-way analysis of variance (ANOVA) with Bonferroni's multiple comparisons test. F (2, 39)=5.816, p=0.0062. n = 14 neurons from four mice for each group. (**D**) Representative images of NAc slices from Chat-Mecp2$^{-/y}$ mice injected with AAV/GFP or AAV/RNAi. (**E**) Statistically integrated immunofluorescence data of α2-GABA$_A$ receptors normalized to AAV/GFP levels. *P*-values were calculated by two-tailed unpaired *t*-test. t = 2.985, df = 39, p=0.0049. n = 27 neurons from three mice (AAV/RNAi), 14 neurons from three mice (AAV/Sham). (**F**) Left: Schematic of cannula implantation. Right: Infusion sites of drugs, stained with DAPI. (**G, H**) Fear deficit was measured by retrieval process in fear conditioning test 20 min after drug application. *P*-values were calculated by two-tailed unpaired *t*-test. t = 2.542, df = 23, p=0.0182 for (**E**). t = 3.609, df = 23, p=0.0015 for (**F**). n = 14 mice for ACSF, n = 11 mice for L-838,417. Data are means ± SEM. *p<0.05, **p<0.01. The online version of this article includes the following source data and figure supplement(s) for figure 6:

**Source data 1.** Statistical reporting of *Figure 6*.

**Figure supplement 1.** Knockdown of α2-GABAA Receptors in Cholinergic Interneurons Did Not Affect mIPSC Frequency in Cholinergic Interneurons, Related to *Figure 6*.

**Figure supplement 2.** Retrograde tracing from NAc cholinergic neurons.

show cue-related fear memory deficits (*Ito-Ishida et al., 2015*; *Adachi et al., 2009*; *Gemelli et al., 2006*). Given that these observations of distinct behaviors manifesting in *Mecp2*-conditioned knockout mice can be attributed to different neuron types located in different brain regions, we proposed that conditional deletion of *Mecp2* from specific neurons in specific brain regions will help dissociate a broad spectrum of RTT phenotypes. To be noted, we used male mice throughout this study. Considering that *Mecp2* gene is located on the X chromosome, female mice are typically mosaic at the cellular level for MeCP2 expression, which may contribute to phenotype variability. Thus, our results with males may not pertain to females.

The ability to associate neutral environmental factors with aversive stimuli is of pivotal importance for survival, and perturbations of associative fear learning may underlie a wide variety of psychiatric disorders (*Adolphs, 2013*; *Tovote et al., 2015*). The amygdala has been emphasized as an essential component in the network that encodes fear (*Adolphs, 2013*; *Johansen et al., 2010*; *Maren and Quirk, 2004*; *Paton et al., 2006*; *Reijmers et al., 2007*; *Janak and Tye, 2015*; *Yu et al., 2017*; *Li et al., 2013*). Furthermore, evidence on the functions and molecular mechanisms of extra-amygdala brain regions regulating fear learning has recently emerged (*Dong et al., 2019*; *Groessl et al., 2018*; *Zhang et al., 2016a*). Nevertheless, whether the NAc, a reward and aversion center in the brain, participates in fear has not been clearly ascertained (*Floresco, 2015*; *Hu, 2016*; *Reynolds and Berridge, 2008*; *Russo and Nestler, 2013*). Here, our evidence indicates that NAc cholinergic interneurons are important to fear learning regardless of whether context or tone was tested. The most probable explanation could be that NAc CIN receives converging input from both HPC and BLA, thus participating in both context and tone fear regulation (*Figure 6—figure supplement 2*). In particular, the function of BLA→NAc circuit has been well reported in tone fear (*Correia et al., 2016*) and HPC→NAc circuit in contextual reward behavior (*LeGates et al., 2018*), indicating the involvement of NAc in both contextual and tone related behaviors. However, further study is necessary to dissect the function of the cholinergic neurons within BLA→NAc and HPC→NAc circuits.

Zhang et al. revealed that activating cholinergic neurons decreases fear response (*Zhang et al., 2016a*). This at first seems contradictory to our findings on the role of NAc CINs in promoting fear. However, there are differences in manipulation (ablation and activation with naïve mice vs. activation with knockout mice), which might contribute to different conclusion. Besides, habenular cholinergic neurons contribute to fear memory expression rather than fear encoding. Our study revealed the role of NAc CINs in fear encoding. Importantly, since these cholinergic neurons are in different brain regions, it is possible that different circuits they are involved and different molecular they express contributed to distinct roles in fear.

The NAc is known to be functionally heterogenous with respect to the regulation of positive and negative affect. Besides, a recent study also suggested the functional difference of CIN in NAc core and shell, with respect to depressive behaviors (*Cheng et al., 2019*). Thus, we got curious about the role of CIN within different NAc subregions in fear regulation. According to our result, inhibition of CIN in core and shell has similar effect on tone/context fear, with a slight difference in NAc medial shell, suggesting a potential heterogeneity within NAc shell. Since the anatomical and characteristics of CINs within shell and core are unknown, future studies are necessary in order to compare CINs in different NAc subregions.

NAc is a crucial brain region for emotional valence (*Hu, 2016*). Our pain threshold test suggested normal pain perception ability with conditional knockout mice, which excluded the possibility that MeCP2 in NAc CINs regulate overall emotion. Besides, our present data suggested that MeCP2 in NAc CINs regulate tone and contextual fear conditioning, both of which are conditioned fear. Importantly, conditioned and innate fear responses seem to be mediated, at least partially, through non-overlapping circuits (*Gross and Canteras, 2012*), making it impossible to simply transfer our understanding of NAc CINs in conditioned fear to others. However, we believe it would be interesting to investigate their role in innate fear.

Cholinergic interneurons of the NAc are tonic firing interneurons that provide rich local innervation to MSNs (*Higley et al., 2009*), however, how they participate in fear learning is unclear. Previous studies exploring the causal relationship between striatal cholinergic interneurons and behaviors have relied on the complete ablation of cholinergic interneurons, which can cause reward-related behavioral deficits (*Kitabatake et al., 2003*; *Xu et al., 2015*). Accumulating evidence suggests that NAc cholinergic interneurons modulate their firing following exposure to cues associated with reward (*Brown et al., 2012*; *Witten et al., 2010*). Furthermore, reduced spontaneous activity of cholinergic interneurons can lead to depressive-like behaviors (*Cheng et al., 2019*). Here, we detected decreased firing rates of cholinergic interneurons in the NAc of Chat-Mecp2$^{-/y}$ mice, with activation of these neurons improving fear learning. It should be noted that light or pharmacological delivery to the NAc cholinergic interneurons was only conducted in fear learning, and the test of fear memory was conducted without additional manipulations as we speculated that ACh signaling is a requisite component in the learning process (*Jiang et al., 2016*). Thus, our results support the notion that cholinergic firing may be a modulator of associative fear learning. However, these results are

inconsistent with conclusions arising from chronic ablation of these neurons (*Kitabatake et al., 2003*; *Xu et al., 2015*). One possible explanation could be that complete ablation of cholinergic interneurons may lead to an indirect compensatory effect, which may mask the direct effect of subtle variations in firing. As MSNs are the major neuron type and only output in the NAc, further studies are required to determine how cholinergic interneurons regulate MSN activity, further modulating fear learning.

We found that the amplitude of miniature inhibitory inputs was significantly increased in Chat-Mecp2$^{-/y}$ mice. In the brain, major inhibitory transmissions are mediated by the GABA$_A$ receptor, which is a ligand-gated chloride channel composed of two α, two β, and one γ sub-unit. Receptors with different α subunits play distinct roles in physiological and pharmacological functions (*Fritschy and Mohler, 1995*; *Rudolph and Knoflach, 2011*; *Rudolph and Möhler, 2004*; *Vinkers et al., 2012*). However, the distribution pattern of different α subunits within the NAc is largely unknown (*Mitchell et al., 2018*; *Koo et al., 2014*). Here, we showed that NAc cholinergic interneurons were more likely to express detectable α2-GABA$_A$ receptors than α1-GABA$_A$ receptors. However, whether inhibitory inputs through the α2-GABA$_A$ receptors are necessary or sufficient for the regulation of cholinergic firing within the NAc is unclear. Previous electrophysiological studies have suggested that blockade of GABA$_A$ receptors with bicuculline produces no obvious alteration in the firing rate of cholinergic interneurons (*Bennett and Wilson, 1999*). However, bicuculline can target all GABA$_A$ receptors, including those in MSNs. The disinhibition effect through GABAergic neurons may counterbalance the direct inhibitory effect on cholinergic interneurons and thus confuse the result. We partially overcame this challenge by applying L-838,417, which can selectively potentiate α2-GABA$_A$ receptors (*Petrache et al., 2019*; *McKernan et al., 2000*). Surprisingly, we showed that bath application of L-838,417 decreased the spiking rate of cholinergic interneurons in slice recordings, which suggests that the α2-GABA$_A$ receptor may be a regulator of cholinergic interneuron spiking in the NAc. As cholinergic interneurons only contribute about 2% of NAc neurons (*Kreitzer, 2009*; *Castro and Bruchas, 2019*), scattered expression of α2-GABA$_A$ receptors in MSNs may buffer the difference resulting from cholinergic interneurons by western blot analysis (*Mitchell et al., 2018*; *Koo et al., 2014*). Thus, we conducted immunohistochemical experiments to support our western blotting results for GABA$_A$ receptor expression in the NAc (*Figure 3J,K*).

MeCP2 is a transcriptional regulator that binds broadly throughout the genome and regulates the expression of thousands of genes (*Chahrour et al., 2008*; *Cheng and Qiu, 2014*; *Zhao et al., 2013*). Previous microarray analysis of gene expression profiles in *Mecp2-null* and *Mecp2-Tg* mice using hypothalamic RNA revealed dysregulation of the *Gabra2* gene in those mouse models (*Chahrour and Zoghbi, 2007*; *Hogart et al., 2007*; *Samaco et al., 2005*). As α2-containing GABA$_A$-receptor selective drugs do not induce tolerance (*Vinkers et al., 2012*), they provide an attractive molecular target for treatment of fear-related neuropsychiatric disorders. However, further studies are required to investigate how MeCP2 regulates the expression of α2-GABA$_A$ receptors in the NAc.

Taken together, our results highlight the role of MeCP2 in regulating GABA$_A$ α2 receptor function in NAc cholinergic interneurons, suggesting a new cellular and molecular mechanism for *Mecp2* in RTT-like phenotypes, and providing strong evidence that modulation of the NAc may ameliorate fear-related disorders.

## Materials and methods

### Key resources table

| Reagent type (species) or resource | Designation | Source or reference | Identifiers | Additional information |
|---|---|---|---|---|
| Genetic reagent (*M. musculus*) | *Chat$^{Cre}$* | Jackson Laboratory | Jax No.006410 | |
| Genetic reagent (*M. musculus*) | *MeCP2$^{Flox/-}$* | Jackson Laboratory | Jax No.006847 | |
| Antibody | Goat polyclonal anti-ChAT | Millipore | Cat# AB144P | IHC: 1:100 |

*Continued on next page*

*Continued*

| Reagent type (species) or resource | Designation | Source or reference | Identifiers | Additional information |
|---|---|---|---|---|
| Antibody | Rabbit monoclonal anti-MeCP2 | Cell Signaling Technology | Cat#3456 s | IHC: 1:200 |
| Antibody | Mouse monoclonal anti-PV | Swant | Cat#PV235 | IHC: 1:1 000 |
| Antibody | Rabbit polyclonal GABA$_A$ receptor α1 | Millipore | Cat#06–868 | IHC: 1:1 000 WB: 1:500 |
| Antibody | Rabbit polyclonal GABA$_A$ receptor α2 | Synaptic System | Cat#224 103 | IHC: 1:1 000 |
| Antibody | Rabbit polyclonal GABA$_A$ receptor α2 | Abcam | Cat#ab72445 | WB: 1:500 |
| Antibody | Rabbit monoclonal anti-GAPDH | Cell Signaling Technology | Cat#5014S | WB: 1:5000 |

## Mouse strains

To specifically delete *Mecp2* in ChAT-positive cholinergic neurons, Chat-Mecp2$^{-/y}$ mice were generated by breeding female *Mecp2$^{flox/-}$* mice (Jax No.006847) with *Chat$^{Cre}$* (Jax No.006410) heterozygous male mice. Mice were group-housed and maintained under standard housing conditions in a temperature (22–25°C)- and humidity (40%)-controlled animal room under a 12 hr light/dark cycle (8:00 am to 8:00 pm) with ad libitum access to food and water. Only animals with cannula implants were housed separately. All mice belonged to the C57BL/6J strain and were housed 3–5 per cage.

## Behavioral assays

Male mice at 9 to 13 weeks of age were used for all tests, considering that female mice are typically mosaic at the cellular level for MeCP2 expression, which may contribute to phenotype variability. We first evaluated the mice for general health, including body weight and fur appearance. The mice were then transferred to the animal facility one week before the behavioral tests. They were allowed to habituate to the testing room for 30 min before test commencement. All the behavior chambers were cleaned with 75% ethanol, except the chambers in fear retrieval tests. All behavioral tests were carried out blind to genotype with age-matched male littermates.

## Open field test

Spontaneous locomotor activity was assessed for 15 min in an arena (45 × 45 × 45 cm). Experiments were conducted under low-light conditions to minimize anxiety effects. Locomotor activity was evaluated as the total distance traveled. Immobility time was evaluated as the time that mice showed the absence of movement over 2000 ms by ANYmaze software.

## Fear conditioning test

The fear conditioning test was performed as described previously, with some modifications (*Li et al., 2011*). A 30-second-train tone (4000 Hz, 80 db) was used as CS and the electric footshock (0.5 mA, 1.5 ms) was used as US. The conditioning session including four trials. The test for context retrieval was performed in the same chamber for 5 min 22 hr after the training. The tone test was conducted 4 hr later. Mice were placed in another chamber with different context for 3 min as baseline, followed by 3 min with tone stimuli presented. While the conditioning context was cleaned with 75% ethanol, the retrieval context and the chamber used in tone fear test were wiped down with 0.3% acetic acid. All events in the fear conditioning test were programmed, and data were recorded through the MED software (MED Associate Inc). Recorded videos were analyzed by Video Freeze Software (MEF Associate Inc). Freezing was defined as motion index <18 for 1 s as the MED software recommended.

## Pain threshold test

A mouse was placed into the conditioning chamber. Every 30 s, a 2 s footshock with 0.05-mA increments (starting from 0 mA) was presented after habituation for 1 min. Two experimenters without

prior knowledge of genotypes or shock intensities scored the vocalization responses. Mouse behavior in the chamber was videotaped for analysis of first signs for discomfort and jumping.

## Immunohistochemistry

Immunohistochemical experiments were performed following standard procedures (*Zhang et al., 2016b*). Images were taken using confocal microscopy (Nikon A1 and Olympus FV3000 Confocal Microscope). For the primary antibodies, we used antibodies for ChAT (Millipore, cat. no. AB144P, 1:100), MeCP2 (Cell Signaling Technology, cat. no. 3456 s, 1:200), PV (Swant, cat. no. PV235, 1:1 000), GABA$_A$ receptor α1 (Millipore, cat. no. 06–868, 1:1 000), and GABA$_A$ receptor α2 (Synaptic System, cat. no. 224 103, 1:1 000).

## Western blot analysis

We performed western blot experiments as described previously (*Rodriguez-Romaguera et al., 2012*). Brains were quickly dissected and the NAc was homogenized in lysis buffer (Beyotime Biotechnology, China) containing 1 mM protease inhibitor PMSF (Beyotime Biotechnology, China). Protein samples were loaded on 10% acrylamide SDS-PAGE gels and then transferred to nitrocellulose membranes. The membranes were incubated with appropriate primary and secondary antibodies, and then visualized by X-ray film exposure (ECL kit, Thermo Scientific). For primary antibodies, we used antibodies for GABA$_A$ receptor α1 (Millipore, cat. no. 06–868, 1:200), GABA$_A$ receptor α2 (Abcam, cat. no. ab72445, 1:500), and GAPDH (Cell Signaling Technology, cat. no. 5014S, 1:5000).

## Preparation of acute slices and electrophysiology

We anesthetized and decapitated 8–12 week-old mice and prepared 350 µm coronal slices using a Vibroslice (Leica VT 1000S, USA) in ice-cold artificial cerebrospinal fluid (ACSF) (composition in mM: 125 NaCl, 3 KCl, 1.25 NaH2PO4, 2 MgSO4, 2 CaCl2, 25 NaHCO3, 10 glucose). The slices were recovered in ACSF at 33°C for 30 min and then kept at room temperature until recording. All solutions were saturated with 95% $O_2$ and 5% $CO_2$. Acute slices were transferred to a recording chamber and fully submerged in continuously perfused (~2 ml/min) ACSF at 25°C. Fluorescent neurons were visually identified under an upright microscope (Nikon, Eclipse FN1) equipped with an infrared-sensitive CCD camera and ×40 water-immersion lens (Nikon, Eclipse FN1). To measure the spontaneous spiking rate of cholinergic interneurons, neurons were patched in a cell-attached configuration and then in whole-cell configuration with electrodes containing the following (in mM): 130 potassium gluconate, 20 KCl, 10 HEPES buffer, 4 Mg-ATP, 0.3 Na-GTP, 10 disodium phosphocreatine and 0.2 EGTA, pH 7.25 with KOH, 288 mOsm. In the whole-cell mode, spontaneous firing was recorded at resting membrane potential. If neurons were not spontaneously active, we discarded the recording. Membrane time constant (Tau) was measured with a single exponential fit of the voltage deflection produced by a small hyperpolarizing current injection from the holding potential (−70 mV). The shape parameters were measured from action potentials by 500 ms current injection and analyzed with MATLAB (MathWorks). Most analyses were performed in Clampfit v10.5 (Axon Instruments, USA). For sIPSC recordings, neurons were clamped in the presence of DNQX (20 µM), and AP5 (50 µM). Microelectrodes were filled with a solution containing 120 mM CsCl, 20 mM Cs-methanesulfonate, 5 mM NaCl, 1 mM MgCl2.6H2O, 10 mM HEPES, 0.2 mM EGTA, 2 mM MgATP, 0.5 mM NaGTP, 0.5 mM spermine, and 5 mM QX314 Chloride; the pH was adjusted to 7.25 with 10 M CsOH. For mIPSC recordings, neurons were clamped at −70 mV in the presence of TTX (1 µM), DNQX (20 µM), and AP5 (50 µM). The intracellular solution contained CsCl 130 mM, NaCl 4 mM, TEA 10 mM, HEPES 10 mM, Na$_2$-ATP 2 mM, Na$_3$-GTP 0.5 mM, and EGTA 0.2 mM. For mEPSC recordings, neurons were clamped at −70 mV in the presence of TTX (1 µM) and picrotoxin (100 µM). We used 130 mM CsMeSO$_3$ to replace CsCl. The synaptic current was then analyzed with Mini60 (Synaptosoft Inc). Photostimulation was applied using a 473 nm laser coupled to an optical fiber controlled by a laser driver (DPSSL II) and digital commands from the Digidata 1440A. Light power at the specimen was ~20 mW/mm$^2$. Light was delivered constantly at 8 Hz for Chat-Mecp2$^{-/y}$ mice. Pipette resistance ranged from 3 to 5 MΩ. Recordings were Bessel-filtered at 10 KHz and sampled at 100 KHz. Access resistance was continuously monitored for each cell. Only neurons with series resistance below 20 MΩ and changing <20% throughout the recording were used for analysis.

## Stereotaxic surgery

The AAV viruses AAV-DIO-ChR2(H134R)-GFP and AAV-DIO-NpHR-mCherry, were purchased from S&E Shanghai Medical Biotechnology Company (China). AAV-DIO-hM3Dq-mCherry, AAV-DIO-hM4Di-mCherry, AAV-DIO-mCherry, and AAV-DIO-GFP were purchased from Shanghai Taitool Bioscience Company (China). AAV-FLEX-Mecp2-GFP was provided by Z.L. Qiu (Institute of Neuroscience, Chinese Academy of Sciences, China).

For α2-GABA$_A$ receptor knockdown, the following short-hairpin sequence was used: 5'-CTGTC TCAGATACAGATATGG-3'. Both the specificity and efficiency of the shRNA were validated and the high titers of engineered AAV-FLEX-α2(RNAi)-GFP and AAV-FLEX-α2(scramble control)-GFP were produced by S&E Shanghai Medical Biotechnology Company (China).

Standard surgical procedures were followed for stereotaxic injection. Briefly, 6–8 week-old male mice were anesthetized with 2% isoflurane and mounted on a custom-built mouse stereotaxic device for surgery. Heart rate and body temperature were continuously monitored. Eye drops were applied to prevent drying. The animals received stereotaxic injections of AAV to the NAc or MS. Coordinates used for MS injection were AP + 0.98 mm and DV −4.5 mm (150 nl) and for NAc injection were AP + 1.1 mm, ML ±1.3 mm, and DV −4.5 mm (200 nl per side). For NAc subregions, we injected 80 nl of virus per side (NAc core: AP + 1.1 mm, ML ±1.3 mm, and DV −4.5 mm; NAc medial shell: AP + 1.1 mm, ML ±0.5 mm, and DV −4.35 mm; NAc ventral medial shell: AP + 1.1 mm, ML ±0.5 mm, and DV −4.75 mm). The AAV vector was delivered with a glass microelectrode into the target coordinates over a 10 min period. Three weeks after the introduction of the viral vector, behavioral tests were performed. A double-cannula with CC 2.6 mm (RWD Life Science, China) was implanted into the bilateral NAc (AP + 1.1 mm, ML ±1.3 mm, and DV −4.4 mm). For each mouse, the cannula was secured using dental cement. After a week of recovery, the mice were subjected to behavioral tests.

## In vivo optogenetic and chemogenetic manipulations

For in vivo optogenetic manipulation, an optical fiber cannula (length 5 mm, NA = 0.22; Inper Inc) was bilaterally implanted into the NAc (AP + 1.1 mm, ML ±1.3 mm, and DV −4.3 mm) and secured using dental cement during the same surgery procedure as for the viral injection. During the behavioral tests, the optical fibers were connected to a laser source using an optical fiber sleeve (Inper Inc). We applied a 5 ms, 8 Hz, continuous 470 nm blue laser stimulation for NAc cholinergic interneurons or continuous 590 nm yellow laser stimulation during the training period of fear conditioning. For chemogenetic manipulation, three weeks after the introduction of AAV-DIO-hM3Dq-mCherry or AAV-DIO-hM4Di-mCherry, CNO (Sigma) was injected intraperitoneally 30 min before conditioning at a dose of 1 mg kg$^{-1}$.

## Quantification and statistical analyses

All data were analyzed using commercially available statistical software packages (Prism five and SPSS v17.0). Sample size was estimated using power analysis based on our preliminary experiments. Data distribution in each experiment was checked for normality before further analysis. Statistical differences were calculated using unpaired two-tailed $t$-tests when comparing two groups or one-way analysis of variance (ANOVA) with Bonferroni's multiple comparison test when comparing three groups with one factor or two-way ANOVA with Tukey's multiple comparison test when constructing learning curves. For comparison of pre- and post-medicine or light application, paired-sample $t$-tests were used. Statistical significance was defined as p<0.05.

## Acknowledgements

We are grateful to ZL Qiu (Institute of Neuroscience, Chinese Academy of Sciences) for providing the AAV/Mecp2 plasmid. We thank GP Feng (Massachusetts Institute of Technology, USA) and Kathleen Quast (Massachusetts Institute of Technology, USA) for insightful comments and editing of the manuscript. We are also grateful to the Core Facilities of Zhejiang University Institute of Neuroscience for technical assistance. This work was supported by the National Key Research and Development Plan of Ministry of Science and Technology of China (2016YF051000), Key Project of the National Natural Science Foundation of China (31430034), Science and Technology Program of Guangdong (2018B030334001), Key Realm R and D Program of Guangdong Province

(2019B030335001), Funds for Creative Research Groups of China from the National Natural Science Foundation of China (81521062), and Non-Profit Central Research Institute Fund of the Chinese Academy of Medical Sciences (2019PT310023) to XM Li.

## Additional information

### Funding

| Funder | Grant reference number | Author |
|---|---|---|
| Chinese Ministry of Science and Technology | National Key Research and Development Plan 2016YF051000 | Xiao-Ming Li |
| National Natural Science Foundation of China | 31430034 | Xiao-Ming Li |
| Guangdong Science and Technology Department | 2018B030334001 | Xiao-Ming Li |
| National Natural Science Foundation of China | Funds for Creative Research Groups 81521062 | Xiao-Ming Li |
| Chinese Academy of Medical Sciences | Non-Profit Central Research Institute Fund 2019PT310023 | Xiao-Ming Li |
| Government of Guangdong Province | Key Realm R&D Program of Guangdong Province 2019B030335001 | Xiao-Ming Li |

The funders had no role in study design, data collection and interpretation, or the decision to submit the work for publication.

### Author contributions

Ying Zhang, Conceptualization, Resources, Data curation, Software, Supervision, Validation, Investigation, Methodology, Writing - original draft, Project administration, Writing - review and editing; Yi Zhu, Data curation, Software, Validation, Investigation, Methodology, Project administration, Writing - review and editing; Shu-Xia Cao, Data curation, Validation, Investigation, Methodology, Project administration, Writing - review and editing; Peng Sun, Data curation, Investigation, Methodology, Writing - original draft, Project administration; Jian-Ming Yang, Investigation, Methodology; Yan-Fang Xia, Data curation, Software, Investigation, Project administration; Shi-Ze Xie, Investigation, Project administration; Xiao-Dan Yu, Jia-Yu Fu, Chen-Jie Shen, Data curation, Software, Methodology; Hai-Yang He, Data curation, Investigation; Hao-Qi Pan, Investigation; Xiao-Juan Chen, Resources, Investigation, Methodology; Hao Wang, Writing - review and editing; Xiao-Ming Li, Conceptualization, Resources, Supervision, Funding acquisition, Validation, Investigation, Visualization, Methodology, Project administration, Writing - review and editing

### Author ORCIDs

Yi Zhu (iD) https://orcid.org/0000-0002-4273-0955
Xiao-Ming Li (iD) https://orcid.org/0000-0002-8617-1702

### Ethics

Animal experimentation: Mouse care and use followed the guidelines of the Animal Advisory Committee at Zhejiang University and the US National Institutes of Health Guidelines for the Care and Use of Laboratory Animals. The care and use of the mice in this work were reviewed and approved by the Animal Advisory Committee at Zhejiang University (ZJU201553001). Every effort was made to minimize suffering.

### Decision letter and Author response

Decision letter https://doi.org/10.7554/eLife.55342.sa1
Author response https://doi.org/10.7554/eLife.55342.sa2

## Additional files

### Supplementary files

• Transparent reporting form

### Data availability

All data generated or analysed during this study are included in the manuscript and supporting files. Source data files have been provided for Figures 1, 2, 3, 4, 5 and 6.

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
