## [Decision Letter]

**Acceptance summary:**

This study investigates a role for a DNA methyl binding protein in fear learning behavior. The study uses a set of elegant experiments to reduce this protein, alter activity, and alter inhibitory receptor function in acetylcholine expressing neurons in a brain region involved with emotional behavior to provide mechanistic insight into fear learning processes. This study has implications for potential mechanisms and brain cell subtypes relevant to fear-related disorders.

**Decision letter after peer review:**

Thank you for submitting your article "MeCP2 in cholinergic interneurons of nucleus accumbens regulates fear memory" for consideration by *eLife*. Your article has been reviewed by three peer reviewers, and the evaluation has been overseen by a Reviewing Editor and Kate Wassum as the Senior Editor. The reviewers have opted to remain anonymous.

The reviewers have discussed the reviews with one another and the Reviewing Editor has drafted this decision to help you prepare a revised submission.

Summary:

This study investigates a role for Mecp2 gene deficiency in cholinergic interneurons (CINs) of the nucleus accumbens (NAc) in fear learning. Using ChAT-Mecp2 deficient mice the authors demonstrate deficits in fear memory retrieval, which was partially restored by expressing Mecp2 to NAc CINs. These effects could be mediated through reduced spontaneous activity in NAc CINs, which occurs through enhanced alpha2-GABAA expression. The authors then use a set of elegant experiments to alter activity and/or knock down alpha2-GABAA in NAc CINs in ChAT-Mecp2 or wildtype mice to further provide a mechanistic role into the fear response deficits occurring with Mecp2 deletion in ChAT neurons. Overall these studies provide a new role for Mecp2 in NAc CINs in fear learning and memory processes.

Essential revisions:

1) A major concern with this study is that the optogenetic, chemogenetic, RNAi, and L-838,417 manipulations are performed during fear learning. To support the conclusion that fear memory is disrupted then these manipulations should be applied during the memory retrieval. Such studies would likely take longer than two months. It is recommended by the reviewers that the authors alter their conclusions to support deficits in fear learning rather than fear memory based on the experiments performed.

2) In line with the above comment, in the data in Figure 1F it appears that the Chat-Mecp2 learning is impaired toward the end of training. Is there a significant difference if Chat-Mecp2 mice are compared to the average of the other 3 groups on day/session/presentation 4? Does the “fear memory” difference between groups occur if freezing is calculated against baseline (e.g. as a percentage of baseline responding or similar)? It's commendable that the authors have performed locomotor and spatial memory controls, but if the effects on test are a result of a more pervasive effect on fear regulation, it would simply change the interpretation to suggest that loss of Mecp2 in CINs results in fear learning deficits.

3) Please provide the data for fear learning in Figure 2C-F and Figure 4.

4) The fact that fear memory was affected by NAc CIN manipulations regardless of whether context or fear was tested is important (e.g. Figures 2C-D) and wasn't given enough attention. This is different from what is observed in other regions that are upstream from NAc, such as basolateral amygdala effects which tend to be stimulus-specific, and hippocampal effects which tend to be context-specific. The data in Figure 6C suggests that different NAc subregions might regulate the stimulus/context fear slightly differently, but without any discussion of these findings in the main body of the paper, it's difficult interpret these findings. Could the authors please comment on this? Is it possible that MeCP2 in NAc CINs regulates all types of fear or overall emotion? The pain threshold control tests argue against the latter, thus please address this.

5) The electrophysiology studies lack sufficient animal numbers. For example, 9 neurons from 2 mice; 7 neurons from 2 mice; n = 14 neurons for ACSF, n = 13 neurons for L-838417 (Figure 3 legend), etc. Please increase the animal numbers in each group.

6) In Figure 3 L-M, please compare the inhibitory effect of L-838,417 on the firing and GABAergic currents of both wild type mice and Chat-Mecp2^-/y^ mice.

7) Please rationalize why the experiments were performed in the BF as well as NAc but not medial habenula (MHb) or other cholinergic rich regions. Please also discuss studies about MHb cholinergic neurons in meditating fear responses and how this contrasts to the current studies. Previous work shows that cholinergic projection neurons of the MHb regulate fear memory expression. Activating MHb cholinergic neurons decreases fear response, which indicates that cholinergic neurons in MHb play an inhibitory role (Zhang et al., 2016). This contrasts to the current study showing the cholinergic interneurons in NAc have an excitatory role since chemogenetic and optogenetic stimulation reverses the reduced firing rate and impaired fear memory by deletion of MeCP2 (Figure 4).

8) It is difficult to distinguish immobility and freezing behaviors using automated software. But, freezing is an active fear reaction, while immobility is an absence of movement. Have the authors measured the freezing behaviors as active fear reactions in the open field test? If so, please describe the experimental procedures with details. If not, please describe procedures with appropriate words such as immobility, motor defects, etc.

9) Please rationalize why the α subunits of GABAA receptor were examined. Are GABAA α1 or α2 Receptor subunits target genes of MECP2 or are they enriched in the CINs?

10) The authors discuss a study showing that knock-down of MECP2 in cholinergic interneurons did not show impairment of fear memory and claimed that the inconsistency results from difference in experimental protocols and in cell-types. Was there an age-difference between the two studies? If so please include this in the text.

11) Please provide the numbers of neurons and/or animals used in Figures 3—figure supplement 1, 2, Figure 4—figure supplement 1, and Figure 6—figure supplement 1? In addition, the number of animals in Figure S6 is low and additional animals would support the conclusions. Further, subsection “Inhibition of NAc Cholinergic Interneurons Resulted in Fear Memory Impairment” proposes to examine CINs in different NAC regions (i.e. core, medial shell and ventromedial shell) to ascertain if they differentially regulate fear memory. However, since the data is in supplemental material there is no information about this outcome this in the text. It is suggested that the data in supplemental figure 6 be added to Figure 5 and that a discussion of these results be included in the main body of the paper. Even if the data are mostly null effects this is would be interesting to know.

12) In Figures 6h-6k, the authors calculated p-values with one-way ANOVA with Dunnett's multiple comparisons and additionally performed two-tailed unpaired t-test. Please explain the reason. It would be better to perform a one-way ANOVA with post-hoc analysis.

---

## [Author Response]

Essential revisions:1) A major concern with this study is that the optogenetic, chemogenetic, RNAi, and L-838,417 manipulations are performed during fear learning. To support the conclusion that fear memory is disrupted then these manipulations should be applied during the memory retrieval. Such studies would likely take longer than two months. It is recommended by the reviewers that the authors alter their conclusions to support deficits in fear learning rather than fear memory based on the experiments performed.

This is a really important issue. Considering that all manipulations were applied during fear learning, rather than retrieval, we agree the conclusion of fear memory deficits cannot be drawn based on the existing data. Accordingly, we have revised our conclusion to fear learning deficits throughout manuscript. We sincerely thank the reviewer for this constructive suggestion.

2) In line with the above comment, in the data in Figure 1F it appears that the Chat-Mecp2 learning is impaired toward the end of training. Is there a significant difference if Chat-Mecp2 mice are compared to the average of the other 3 groups on day/session/presentation 4? Does the “fear memory” difference between groups occur if freezing is calculated against baseline (e.g. as a percentage of baseline responding or similar)? It's commendable that the authors have performed locomotor and spatial memory controls, but if the effects on test are a result of a more pervasive effect on fear regulation, it would simply change the interpretation to suggest that loss of Mecp2 in CINs results in fear learning deficits.

The reviewers are correct. Compared to the other 3 control groups, freezing time of Chat-Mecp2 group is significantly decreased on trial 4 (Figure 1—figure supplement 1A). Besides, if freezing is calculated against trial 4 during learning, the “fear memory deficits” with Chat-Mecp2 group disappeared (Figure 1—figure supplement 1B, C), suggesting that the impairment on retrieval day could attribute to fear learning deficits. We thank the reviewers for pointing out this issue. The new data and figures have been added and the interpretation has been revised.

3) Please provide the data for fear learning in Figure 2C-F and Figure 4.

These data have now been included in the manuscript (Figure 2—figure supplement 1 and Figure 4—figure supplement 1D, F).

Figure 2—figure supplement 1

“To test the behavioral effects of MeCP2 restoration in cholinergic neurons, we performed fear conditioning tests. It appeared that fear learning deficit of Chat-Mecp2^-/y^ mice was rescued towards the end of training with MeCP2 restoration in the cholinergic neurons of NAc, but not medial septum (MS) (Figure 2—figure supplement 1).”

Figure 4—figure supplement 1

“Activation of NAc cholinergic interneurons significantly increased the freezing level of Chat-Mecp2^-/y^ mice in the training process (Figure 4—figure supplement 1C), as well as the context and tone fear memory retrieval process (Figure 4B, C).”

“We found that optogenetic activation of cholinergic interneurons (8 Hz) only during fear learning also rescued fear deficits in Chat-Mecp2^-/y^ mice (Figure 4E, F and Figure 4—figure supplement 1E).”

4) The fact that fear memory was affected by NAc CIN manipulations regardless of whether context or fear was tested is important (e.g. Figures 2C-D) and wasn't given enough attention. This is different from what is observed in other regions that are upstream from NAc, such as basolateral amygdala effects which tend to be stimulus-specific, and hippocampal effects which tend to be context-specific. The data in Figure 6C suggests that different NAc subregions might regulate the stimulus/context fear slightly differently, but without any discussion of these findings in the main body of the paper, it's difficult interpret these findings. Could the authors please comment on this? Is it possible that MeCP2 in NAc CINs regulates all types of fear or overall emotion? The pain threshold control tests argue against the latter, thus please address this.

We thank the reviewers for raising these important discussions which have been missing in our original manuscript. As mentioned by the reviewers, other brain regions that are affected in fear regulation always tend to be either context-specific or stimulus-specific. For example, hippocampal lesion produces a severe deficit in the acquired context fear, but not tone fear (Kim et al., 1992, Science). Lesions of the amygdala interfered with the conditioning of fear responses of both the context and tone, whereas its subregion, BLA and LA differentially contribute to context and tone-US association (Calandreau et al., 2005, Learning and Memory). Based on our data, NAc manipulation resulted in alteration of both.

The most probable explanation could be that NAc CIN receives converging input from both HPC and BLA, thus participating in both context and tone fear regulation (Figure 6—figure supplement 2). In particular, the function of BLANAc circuit has been well reported in tone fear behavior (Correia et al., 2016) and HPCNAc circuit in contextual reward behavior (LeGates et al., 2018), indicating the involvement of NAc in both contextual and tone related behaviors.

The NAc is known to be functionally heterogenous with respect to the regulation of positive and negative affect (Hamel et al., 2017; Ito and Hayen et al., 2011). Besides, a recent study also suggested the functional difference of CIN in NAc core and shell, with respect to depressive behaviors (Cheng et al., 2019). Thus, we got curious about the role of CIN within different NAc subregions in fear regulation. According to our result (Figure 5), inhibition of CIN in core and shell have similar effect on tone/context fear, with a slight difference in NAc medial shell. Inhibition of CINs in NAc medial shell impaired tone fear, with context fear intact. This result suggested a potential heterogeneity within NAc shell. Since the anatomical and characteristics of CINs within shell and core are unknown, future studies are necessary in order to compare CINs in different NAc subregions.

NAc is a crucial brain region for emotional valence (Hu, 2016). Our present data suggested that MeCP2 in NAc CINs regulate tone and contextual fear conditioning, both of which are conditioned fear. Importantly, conditioned and innate fear responses seem to be mediated, at least partially, through nonoverlapping circuits (Gross and Canteras, 2012), making it impossible to simply transfer our understanding of NAc CINs in conditioned fear to others. However, we believe it would be interesting to investigate their role in innate fear.

As indicated by the reviewers, our pain threshold test suggested normal pain perception ability with conditional knockout mice, which excluded the possibility that MeCP2 in NAc CINs regulate overall emotion.

We discussed this issue in the revised manuscript in the Discussion session.

5) The electrophysiology studies lack sufficient animal numbers. For example, 9 neurons from 2 mice; 7 neurons from 2 mice; n = 14 neurons for ACSF, n = 13 neurons for L-838417 (Figure 3 legend), etc. Please increase the animal numbers in each group.

We apologize for not stating clearly and lack of sufficient animal numbers for some groups. All electrophysiology groups included neurons from at least 3 animals in the revised manuscript.

6) In Figure 3 L-M, please compare the inhibitory effect of L-838,417 on the firing and GABAergic currents of both wild type mice and Chat-Mecp2^-/y^ mice.

We thank the reviewers for pointing out this important issue. We found potentiation of α2-GABAA receptors with L-838,417 significantly decreased the firing frequency of NAc CHIs in Chat-Mecp2^-/y^ mice (Figure 3—figure supplement 2D). Furthermore, it increased both the frequency and amplitude of sIPSC in NAc slices from both control and Chat-Mecp2^-/y^ mice (Figure 3—figure supplement 2E, F). To be noted, L-838-417 has been reported to increase both the frequency and amplitude of sIPSC with Alzheimer’s disease mouse model (Petrache et al., 2019). As suggested, we have added this part into current manuscript.

Figure 3—figure supplement 2

“Potentiating the α2-GABAA receptors with L-838,417 (2 μM) in bath solution significantly decreased the firing frequency of cholinergic interneurons in NAc slices from both control (Figure 3L, M) and Chat-Mecp2^-/y^ mice (Figure 3—figure supplement 2D). Further, we also compared the effect of L-838,417 on GABAergic current and found that both the frequency and amplitude of sIPSC have been increased after treatment (Figure 3—figure supplement 2E, F).”

7) Please rationalize why the experiments were performed in the BF as well as NAc but not medial habenula (MHb) or other cholinergic rich regions. Please also discuss studies about MHb cholinergic neurons in meditating fear responses and how this contrasts to the current studies. Previous work shows that cholinergic projection neurons of the MHb regulate fear memory expression. Activating MHb cholinergic neurons decreases fear response, which indicates that cholinergic neurons in MHb play an inhibitory role (Zhang et al., 2016). This contrasts to the current study showing the cholinergic interneurons in NAc have an excitatory role since chemogenetic and optogenetic stimulation reverses the reduced firing rate and impaired fear memory by deletion of MeCP2 (Figure 4).

We thank the reviewers for these important questions.

Our previous work (Zhang et al., 2016) suggested that conditional deletion of MECP2 in cholinergic neurons caused several specific RTT phenotypes. Re-expressing MECP2 in basal forebrain cholinergic neurons could rescue most except fear encoding. We thus initiated this study to expand on the knowledge (Year 2014): what cholinergic brain region is responsible for fear encoding deficit with Chat-Mecp2^-/y^ mice? Both NAc and MHb have been reported to be important in mood regulation. However, among all cholinergic nuclei, NAc is the only one that connect directly with both hippocampus and amygdala, two well recognized structures that regulate fear encoding. This suggested the potential role of NAc CINs in fear encoding.

Zhang et al. (Zhang., 2016) revealed that activating cholinergic neurons decreases fear response. This at first seems contradictory to our findings on the role of NAc CINs in promoting fear. However, there are differences in manipulation (ablation and activation with naïve mice vs. activation with knockout mice), which might contribute to different conclusion. Besides, habenular cholinergic neurons contribute to fear memory expression rather than fear encoding. Thanks to reminding from the reviewers, our study revealed the role of NAc CINs in fear encoding. Importantly, since these cholinergic neurons are in different brain regions, it is possible that different circuits they are involved and different molecular they express contributed to distinct roles in fear behavior.

We discussed this issue in the revised manuscript in the Discussion session.

8) It is difficult to distinguish immobility and freezing behaviors using automated software. But, freezing is an active fear reaction, while immobility is an absence of movement. Have the authors measured the freezing behaviors as active fear reactions in the open field test? If so, please describe the experimental procedures with details. If not, please describe procedures with appropriate words such as immobility, motor defects, etc.

We did not measure the freezing behaviors in open field test. Thus, we agree it is inaccurate to use the word “freezing” without appropriate description. To make it clearer, we added the procedure in the Materials and methods part: “Recorded videos were analyzed by Video Freeze Software (MEF Associate Inc). Freezing was defined as motion index < 18 for 1s as the MED software recommended.”

9) Please rationalize why the α subunits of GABAA receptor were examined. Are GABAA α1 or α2 Receptor subunits target genes of MECP2 or are they enriched in the CINs?

We thank the reviewers for pointing this out. Exactly like the reviewers’ predictions, GABAA receptor is a ligand-gated chloride channel composed of two α, two β, and one γ subunits. According to the in-situ data from Allen Brain, β3 subunit is widely expressed in NAc, while α2 is sparsely expressed with a pattern similar to cholinergic neurons. Based on this, we then investigated the specific expression of α2 at CINs with α1 as control.

10) The authors discuss a study showing that knock-down of MECP2 in cholinergic interneurons did not show impairment of fear memory and claimed that the inconsistency results from difference in experimental protocols and in cell-types. Was there an age-difference between the two studies? If so please include this in the text.

We thank the reviewers for pointing out this possibility. There is age difference between these two studies. Ballinger et al. conducted fear conditioning test when mice were 21 weeks old. However, the age of mice we used throughout experiments were between 9 to 12 weeks. Since Rett Syndrome is a neurodevelopmental disorder, there is high possibility that age difference of mice between two studies lead to distinct results. This part has been added into the manuscript in paragraph two of the Discussion.

11) Please provide the numbers of neurons and/or animals used in Figures 3—figure supplement 1, 2, Figure 4—figure supplement 1, and Figure 6—figure supplement 1? In addition, the number of animals in Figure S6 is low and additional animals would support the conclusions. Further, subsection “Inhibition of NAc Cholinergic Interneurons Resulted in Fear Memory Impairment” proposes to examine CINs in different NAC regions (i.e. core, medial shell and ventromedial shell) to ascertain if they differentially regulate fear memory. However, since the data is in supplemental material there is no information about this outcome this in the text. It is suggested that the data in supplemental figure 6 be added to Figure 5 and that a discussion of these results be included in the main body of the paper. Even if the data are mostly null effects this is would be interesting to know.

We apologize for lack of information on neuron/animal numbers and sufficient animals in Figure S6. We agree that it is hard to appreciate the figure since there was no description about its result in the manuscript. Thus, we have moved Figure S6 into Figure 5 as suggested and corresponding discussion has been included in the revised paper. We sincerely thank the reviewers for this constructive suggestion.

12) In Figures 6h-6k, the authors calculated p-values with one-way ANOVA with Dunnett's multiple comparisons and additionally performed two-tailed unpaired t-test. Please explain the reason. It would be better to perform a one-way ANOVA with post-hoc analysis.

We apologize for not stating it clearly. We compared the effect of α2RNAi in three groups (*Chat^Cre^* mice with control virus, Chat-Mecp2^-/y^ mice with control virus, Chat-Mecp2^-/y^ mice with α2RNAi virus) with one-way ANOVA. We applied t-test to address the effect of inhibition in two groups (Chat-Mecp2^-/y^ mice with α2RNAi virus, Chat-Mecp2^-/y^ mice with α2RNAi virus and NpHR virus). We assumed there are two different factors, so compared them separately with different method. Thank you for pointing this out, we have used one-way ANOVA for this figure in our revised manuscript.

**References**

Calandreau, L., Desmedt, A., Decorte, L. & Jaffard, R. 2005, A different recruitment of the laterl and basolateral amygdala promotes contexual or elemental conditioned association in pavlovian fear conditioning. Learning & Memory, 12 (4): 383-8.

Hamel, L., Thangarasa, T., Samadi, O. & Ito, R. 2017. Caudal Nucleus Accumbens Core Is Critical in the Regulation of Cue-Elicited Approach-Avoidance Decisions. eneuro, 4.

Ito, R. & Hayen, A. 2011. Opposing roles of nucleus accumbens core and shell dopamine in the modulation of limbic information processing. J Neurosci, 31, 6001-7.

Kim, J. J. & Fanselow, M. S. 1992. Modality-specific retrograde amnesia of fear. Science, 256: 675-677.